# Structure of a TRPM2 channel in complex with Ca$^{2+}$ explains unique gating regulation

**Zhe Zhang[1,2†]\*, Balázs Tóth[3,4†], Andras Szollosi[3,4], Jue Chen[1,2], László Csanády[3,4]\***

[1]Laboratory of Membrane Biophysics and Biology, The Rockefeller University, New York, United States; [2]Howard Hughes Medical Institute, Chevy Chase, United States; [3]Department of Medical Biochemistry, Semmelweis University, Budapest, Hungary; [4]MTA-SE Ion Channel Research Group, Semmelweis University, Budapest, Hungary

**\*For correspondence:**
zzhang01@mail.rockefeller.edu
(ZZ);
csanady.laszlo@med.semmelweis-univ.hu (LC)

[†]These authors contributed equally to this work

**Abstract** Transient receptor potential melastatin 2 (TRPM2) is a Ca$^{2+}$-permeable cation channel required for immune cell activation, insulin secretion, and body heat control. TRPM2 is activated by cytosolic Ca$^{2+}$, phosphatidyl-inositol-4,5-bisphosphate and ADP ribose. Here, we present the ~3 Å resolution electron cryo-microscopic structure of TRPM2 from *Nematostella vectensis*, 63% similar in sequence to human TRPM2, in the Ca$^{2+}$-bound closed state. Compared to other TRPM channels, TRPM2 exhibits unique structural features that correlate with its function. The pore is larger and more negatively charged, consistent with its high Ca$^{2+}$ selectivity and larger conductance. The intracellular Ca$^{2+}$ binding sites are connected to the pore and cytosol, explaining the unusual dependence of TRPM2 activity on intra- and extracellular Ca$^{2+}$. In addition, the absence of a post-filter motif is likely the cause of the rapid inactivation of human TRPM2. Together, our cryo-EM and electrophysiology studies provide a molecular understanding of the unique gating mechanism of TRPM2.

DOI: https://doi.org/10.7554/eLife.36409.001

## Introduction

TRPM2 belongs to the M (Melastatin) subfamily of transient receptor potential (TRP) ion channels. Despite a high degree of sequence conservation within the subfamily, its eight members are involved in a multitude of biological processes, and are regulated by diverse stimuli (*Kraft and Harteneck, 2005*). TRPM2 plays key roles in migration and chemokine production of immune cells (*Yamamoto et al., 2008*; *Knowles et al., 2011*), insulin secretion (*Uchida et al., 2011*), and body heat control (*Song et al., 2016*; *Tan and McNaughton, 2016*). Under pathological conditions such as stroke and myocardial infarction TRPM2 activity leads to apoptosis (*Nilius et al., 2007*), and genetic linkage studies strongly suggest its involvement in the development of amyotrophic lateral sclerosis, Parkinsonism Dementia (*Hermosura et al., 2008*), and bipolar disorder (*McQuillin et al., 2006*; *Xu et al., 2006*). TRPM2 is therefore an emerging therapeutic target for chronic inflammatory and neurodegenerative diseases, diabetes, and hyperinsulinism.

TRPM2 is a Ca$^{2+}$ permeable non-selective cation channel opened by simultaneous binding of three obligate co-activators, cytosolic ADP-ribose (ADPR) (*Perraud et al., 2001*; *Sano et al., 2001*; *Hara et al., 2002*), Ca$^{2+}$ (*Csanády and Töröcsik, 2009*), and membrane phosphatidylinositol 4,5-bisphosphate (PIP$_2$) (*Tóth and Csanády, 2012*). The location of the binding sites for activating Ca$^{2+}$ are debated (*McHugh et al., 2003*; *Starkus et al., 2007*; *Csanády and Töröcsik, 2009*; *Kühn et al., 2015*), and regulation by Ca$^{2+}$ and PIP$_2$ are interrelated (*Tóth and Csanády, 2012*) through a mechanism which is not understood.

Similarly to other TRPM subfamily channels, homotetrameric TRPM2 channels contain an ~800 residue N-terminal region, a transmembrane domain (TMD), a TRP domain, and a coiled-coil. In TRPM2 a C-terminal ~270 residue NUDT9-homology (NUDT9H) domain serves to bind ADPR (*Perraud et al., 2001*). So far, high-resolution structures are available only for TRPM4 and TRPM8 (*Guo et al., 2017*; *Autzen et al., 2018*; *Winkler et al., 2017*; *Yin et al., 2018*). TRPM2 and TRPM4 are both activated by cytosolic $Ca^{2+}$. However, as TRPM4 is monovalent selective, its activation requires a nearby $Ca^{2+}$ source, such as a co-localized $Ca^{2+}$ channel (*Launay et al., 2002*; *Kraft and Harteneck, 2005*). In contrast, TRPM2 is itself $Ca^{2+}$ permeable, which provides a unique mechanism for positive feedback (*Perraud et al., 2001*; *Csanády and Töröcsik, 2009*). Furthermore, low micromolar ATP, ADP, and AMP inhibit TRPM4 (*Nilius et al., 2004*), but do not affect TRPM2 (*Sano et al., 2001*; *Tóth and Csanády, 2010*; *Moreau et al., 2013*).

Here, we determined the structure of the TRPM2 channel from the cnidarian *Nematostella vectensis* (nvTRPM2) in complex with $Ca^{2+}$ by electron cryo-microscopy (cryo-EM), at a resolution of 3.1 Å, and correlated the structure with detailed biophysical characterization of channel permeation and gating properties. Although the global architecture of nvTRPM2 is similar to that of TRPM4, distinct local structural features provide explanations for permeation and gating properties that are unique to TRPM2, in particular to its regulation by intra- and extracellular $Ca^{2+}$ as well as by $PIP_2$.

## Results and discussion

### Basic functional properties of nvTRPM2

The nvTRPM2 protein shows ~63% sequence similarity (34% sequence identity) with human TRPM2 (hTRPM2), and when expressed in mammalian cells, it forms functional channels opened by cytosolic ADPR (*Kühn et al., 2015*). Because of the limitations associated with controlling and altering cytosolic ligand concentrations in whole-cell recordings, we expressed nvTRPM2 channels in *Xenopus laevis* oocytes, and studied their biophysical properties in inside-out cell-free patches, under rapid continuous superfusion of the cytosolic membrane surface (*Figure 1*, *colored bars*). In the absence of extracellular (pipette) $Ca^{2+}$ ($[Ca^{2+}]_o$ buffered to ~1 nM), large macroscopic nvTRPM2 currents could be activated by superfusion of the cytosolic patch surface with ADPR (*black bars*, 100 µM) and $Ca^{2+}$ (*gray bars*, 40 µM). ADPR and intracellular $Ca^{2+}$ acted as obligate co-activators, as addition of both ligands was required to open, whereas removal of either ligand was sufficient to close the channels (*Figure 1A*). Although nvTRPM2 current elicited by saturating ADPR + $Ca^{2+}$ was only modestly (by 10–20%) enhanced upon exposure to exogenous $PIP_2$ (*Figure 1B*, *left blue bar*), it was almost completely abolished when membrane $PIP_2$ headgroups were masked with polylysine (*Figure 1B*,

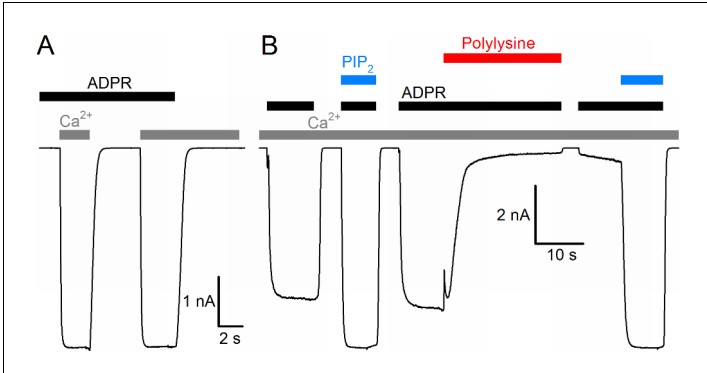

**Figure 1.** Basic functional properties of the *Nematostella vectensis* (nv) TRPM2 channel. (**A**) Macroscopic inward $Na^+$ currents in an inside-out patch excised from a *Xenopus laevis* oocyte overexpressing nvTRPM2, evoked by superfusion of the cytoplasmic patch surface with 100 µM ADPR (*black bar*) plus 40 µM free $Ca^{2+}$ (*gray bars*); extracellular (pipette) $[Ca^{2+}]$ was ~1 nM, membrane potential was −20 mV. (**B**) Effects of 25 µM dioctanoyl-$PIP_2$ (*blue bars*) on nvTRPM2 channel currents before and after exposure to 15 µg/ml polylysine (*red bar*). Conditions as in (**A**).
DOI: https://doi.org/10.7554/eLife.36409.002

*red bar*), but could then be fully recovered by addition of exogenous PIP$_2$ (*Figure 1B*, *right blue bar*). Thus, PIP$_2$ is also essential for nvTRPM2 activity, but is tightly bound, so that the channels remain close-to-saturated even at the low PIP$_2$ concentration of an inside-out patch. All in all, gating regulation by ADPR, Ca$^{2+}$, and PIP$_2$ of nvTRPM2 is very similar to that of hTRPM2 (*Tóth and Csanády, 2012*).

## Structure determination

To understand the unique functional properties of TRPM2, we purified the full-length nvTRPM2 protein from HEK 293 S cells and analyzed its structure by cryo-EM in the presence of trace amounts of Ca$^{2+}$ and absence of ADPR. (*Figure 2—figure supplement 1* and *Figure 2—figure supplement 2*). The two-dimensional (2D) classification analysis showed clear secondary structure features and subsequent 3D classification revealed a single conformation (*Figure 2—figure supplement 1*). The final reconstruction, calculated with 72% of the particles from 3D classification, was refined to 3.1 Å resolution (*Figure 2—figure supplement 1* and *Figure 2—figure supplement 2*). This map showed excellent side chain densities in most of the regions, which allowed us to build ~1200 residues de novo without ambiguity. The C-terminal NUDT9H domain, however, is entirely invisible in the density map, indicating that in the absence of ADPR this domain is flexibly linked to the core structure.

## Overall structure of nvTRPM2

The nvTRPM2 structure is shaped like a square prism with dimensions of 135 x 100 x 100 Å (*Figure 2A–C*, color coded by subunits), and >80% of its volume is cytosolic. The large N-terminal cytosolic region can be subdivided (Fig. 2D) into an α/β N-terminal domain (NTD, *blue*), a small ankyrin repeat domain consisting of two helical hairpins (ARD, *marine*), a linker helix domain (LHD, *cyan*), and a Pre-S1 helix (*orange*). Like in all known TRP channel structures, the first four transmembrane helices assemble into a voltage sensor-like domain (VSLD, S1-S4, *yellow*) and the last two form the pore domain (PD, S5-S6, *green*). TRP helix 1 is an extension of helix S6, and together with TRP helix 2 and an unstructured loop (TRP loop) forms the TRP domain (*red*). At the C terminus a 'stretcher' helix (*pink*) is followed by a coiled-coil helix (CC, *purple*) (*Figure 2D*, see also *Figure 2—figure supplement 3*).

The nvTRPM2 structure is assembled from three layers (*Figure 2E–F*). The bottom tier contains the NTDs, the ARDs, and the CCs. Through extensive interactions of the NTDs with the ARD of both the same (*Figure 2—figure supplement 4A*) and of the adjacent subunit (*Figure 2—figure supplement 4B*), these domains assemble into a square ring (*Figure 2C*; *Figure 2F*, *Bottom tier*) which accommodates the CC helix at its center (*Figure 2C*). Apart from a salt bridge between R154 in the NTD and D1207 in the CC (*Figure 2—figure supplement 4A*) there is little contact between the latter two domains. The middle tier contains the LHDs, the TRP loops, and the stretcher helices (*Figure 2E–F*). Relative to the TRPM4 structures two extra helices (LH13 and LH14, *Figure 2—figure supplement 3* and *Figure 2—figure supplement 4*) are clearly resolved in the LHDs of nvTRPM2. Intersubunit interactions of LH13 with the LHD and N-terminal loop of the adjacent subunit (*Figure 2—figure supplement 4B*) assemble the four LHDs into a second square ring (*Figure 2F*, *Middle tier*; Figure 5A, bottom). Descending from the top layer, the TRP loop embraces the LHD ring from the periphery, and continues into the stretcher helix. Below the LHD ring, intimately connected to the latter, the four stretcher helices converge into the central CC, like the stretchers of an umbrella into its shaft (*Figures 2E* and 5A, *Figure 2—figure supplement 4 pink* and *purple*). Extensive intrasubunit interactions of the LHDs with TRP helix 1 and the pre-S1 helix (*Figure 2—figure supplement 4A*) connect the middle tier to the top tier, whereas the tight turn between the stretcher and CC helices provides a direct link between the middle and bottom tiers. The top tier is formed by the Pre-S1 helices, the transmembrane domain, and TRP helices 1 and 2 (*Figure 2E–F*). The structure of the transmembrane VSLD (S1-S4, *Figure 2E–F*, *yellow*) and PD (S5-S6, *Figure 2E–F*, *green*) resembles that of other voltage-gated cation channels, and the VSLD is domain-swapped relative to the PD (*Figure 2B*). However, the S4 segment contains only two conserved arginines (*Figure 2—figure supplement 3*, *blue boxes*), and extensive hydrophobic interactions between the S4 and S5 helices of adjacent subunits (*Figure 2—figure supplement 4B*), similar to those observed in TRPM4, make large translational movements within the VSLD unlikely. TRP helix 1, which contains the conserved TRP box motif, is a cytosolic extension of helix S6. Due to a sharp bend at the end of S6, TRP helix 1

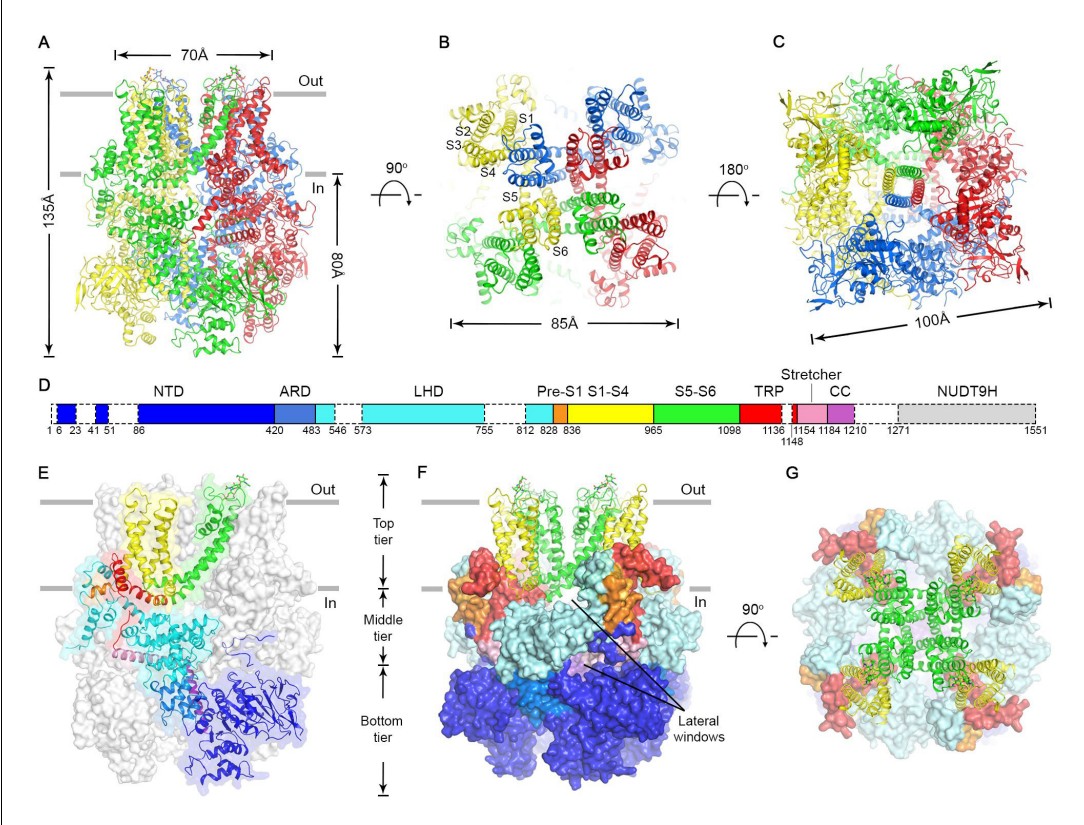

**Figure 2.** Cryo-EM structure of the nvTRPM2 channel. (A–C) Different views of the overall structure of the nvTRPM2 tetramer. Protomers are color coded. Gray bars in (A) represent approximate membrane boundaries. Transmembrane helices S1-S6 of one subunit are labeled in (B). (D) Schematic domain structure of nvTRPM2. Regions not built in the final model are indicated with dashed boxes. (E–G) Domain organization of nvTRPM2. In (E), one subunit is shown in ribbon, the remaining subunits as gray surfaces. In (F) and (G), the transmembrane domains are represented as ribbon, the cytosolic domains as surfaces. Domain color coding in (E–G) follows that in (D). Two N-Acetylglucosamines attached to residue N1017 are shown as sticks. See also *Figure 2—figure supplements 1–5*.

DOI: https://doi.org/10.7554/eLife.36409.003

The following figure supplements are available for figure 2:

**Figure supplement 1.** Cryo-EM structure determination and evaluation.

DOI: https://doi.org/10.7554/eLife.36409.004

**Figure supplement 2.** Validation of the atomic model.

DOI: https://doi.org/10.7554/eLife.36409.005

**Figure supplement 3.** Multiple sequence alignment of TRPM proteins.

DOI: https://doi.org/10.7554/eLife.36409.006

**Figure supplement 4.** Intra- and inter-subunit interactions.

DOI: https://doi.org/10.7554/eLife.36409.007

**Figure supplement 5.** Structural comparisons between individual domains, one subunit, or the entire tetramer of nvTRPM2 and other TRPM family channels.

DOI: https://doi.org/10.7554/eLife.36409.008

runs parallel to the membrane, packed against the cytosolic surface of transmembrane helices S1, S2, and S5 (*Figure 2—figure supplement 4A*), from the center of the protein towards its periphery, where the hydrophobic TRP helix 2 reenters the membrane (*Figure 2E*, *red*). At the periphery of the top layer a tight interaction hub is formed between the short Pre-S1 helix (*Figure 2E*, *orange*) and helices 12 and 13 of the LHD (*Figure 2—figure supplement 4A*).

Consistent with the high sequence conservation within the TRPM subfamily (*Figure 2—figure supplement 3*), the overall architecture of nvTRPM2 (*Figure 2D–E*) resembles that of TRPM4 and TRPM8. However, there are important structural differences that correlate with the different regulatory mechanisms of these channels. For example, in TRPM4, a high-affinity nucleotide binding site

for inhibitory ATP, ADP, and AMP is located at the interface between the NTD and the ARD of the neighboring subunit (*Guo et al., 2017*). In nvTRPM2 the five long intertwined loops that build this surface (*Figure 2—figure supplement 5*, *NTD, right surface* and *Figure 2—figure supplement 4B*, *inset 3*) are differently arranged, contain an extra β strand (β7b, *Figure 2—figure supplement 4B*, *inset 3*), and lack residues important for nucleotide binding, explaining insensitivity of TRPM2 to these nucleotides. Relative domain positioning is also significantly different in nvTRPM2 (*Figure 2—figure supplement 5*). When viewed from the extracellular side, the ring formed by the four NTDs is rotated counterclockwise in nvTRPM2 relative to TRPM4 but clockwise relative to TRPM8 (*Figure 2—figure supplement 5*, *Tetramer*): when the pore domains of the three channels are superimposed, the displacement is 13–18 Å at the level of the NTDs (*Figure 2—figure supplement 5*, *Subunit*). Within the limitations of its lower resolution, the TRPM8 structure seems the least similar to TRPM4 and nvTRPM2, with a substantially different arrangement of its C-terminal helices relative to the transmembrane domain (*Figure 2—figure supplement 5*, *TM-Stretcher-CC*).

## Ion channel pore

The PD is formed by helix S5, the turret loop, the pore helix, the filter, a short post-filter helix, a post-filter loop ('outer pore loop'), and helix S6. Asparagine 1017 in the turret loop is glycosylated, two N-acetyl-glucosamine molecules are clearly visible (*Figure 2E* and *Figure 2—figure supplement 2A*). The selectivity filter is short and wide (*Figure 3A*), the diameter at its narrowest point is ~5.2 Å (*Figure 3B*), larger than that of TRPM4 (~4.2 Å). Such a large diameter is consistent with functional properties of the hTRPM2 pore which permeates tetramethyl-ammonium (~5.8 Å) but not N-methyl-D-glucosamine (~6.8 Å) (*Tóth and Csanády, 2012*), and suggests that the TRPM2 filter allows hydrated ions to pass. The nvTRPM2 filter is lined by the backbone atoms of Y1035 and G1036, and the side chain of E1037 (*Figure 3A*). A sausage-like density is observable in the filter, which probably represents two possible positions of a cation, coordinated by the 1035–1036 peptide carbonyl groups (*Figure 3C*). These two ions, and a third cation in the central cavity, were modeled as $Na^+$, the predominant cation in the buffer. E1037 in nvTRPM2 is replaced by a glutamine in hTRPM2, hTRPM4, and hTRPM5 (*Figure 2—figure supplement 3*, *green box*). This sequence difference might underlie the much larger $Ca^{2+}$ preference of the nvTRPM2 pore. Indeed, the relative permeability for $Ca^{2+}$ versus $Na^+$ ($p_{Ca}/p_{Na}$), estimated from the reversal potential of unitary currents under biionic conditions, is ~35 for nvTRPM2 (*Figure 3G*, *black*), as opposed to ~0.45 for hTRPM2 (*Figure 3G*, *blue*, replotted from [*Tóth and Csanády, 2012*]), and ~0 for hTRPM4 (*Launay et al., 2002*) and hTRPM5 (*Liu and Liman, 2003*).

The external vestibule of nvTRPM2 is lined by a double ring of negative charges (*Figure 3D*), the side chains of D1041 and E1042 of the post-filter helix (LDE motif, *Figure 3H*, *inset*, *red box*), and additional negatively charged side chains (E1046 and E1050) line the upper parts of the vestibule (*Figure 3A*). It is also substantially narrower (diameter 10.0 Å) than that of TRPM4 (16.6 Å, *Figure 3—figure supplement 1B–C*), because the shorter outer pore loop of nvTRPM2 (residues 1045–1055) flips towards the pore (*Figure 3—figure supplement 1A–B*), stabilized by a salt bridge between K1047 and E1042 (*Figure 3A*, *blue side chains*). Crowding of four times four acidic residues in the tight external vestibule results in a larger negative surface charge density in nvTRPM2 compared to TRPM4 (*Figure 3—figure supplement 1C*). The shape and surface electrostatics of the internal vestibule of TRPM family channels is also diverse (*Figure 3—figure supplement 1B,D*): the nvTRPM2 internal vestibule is tighter (diameter 4.7 Å) than that of TRPM4 (8.5–13 Å), and also much more negative, due to the presence of N1086 (*Figure 3A* and *Figure 3—figure supplement 1B*), E1090 (*Figure 3A,E* and *Figure 3—figure supplement 1B*), and E1094 (*Figure 3—figure supplement 1B*). Two of these residues are conserved in hTRPM2, but only one in TRPM8, and none of the three in TRPM4 (*Figure 2—figure supplement 3*, *pink boxes*).

The larger pore diameter and the larger negative surface charge density in both vestibules of nvTRPM2 likely account for its larger throughput rate for cations: in symmetrical 144 mM $Na^+$, at negative voltages, unitary conductance approaches ~ 150 pS for nvTRPM2 (*Figure 3F*, *black*; note block of outward currents by internal 125 µM $Ca^{2+}$), as compared to ~25 pS for TRPM4 (*Launay et al., 2002*). Human TRPM2 also has a large pore diameter (*Tóth and Csanády, 2012*), but lacks the two acidic residues in the post-filter helix (*Figure 3H*, *inset*, *red box*), and one acidic residue in the internal vestibule. These differences likely account for its intermediate conductance (~50 pS in symmetrical 144 mM $Na^+$, *Figure 3F*, *blue*) between TRPM4 and nvTRPM2. Indeed, inserting

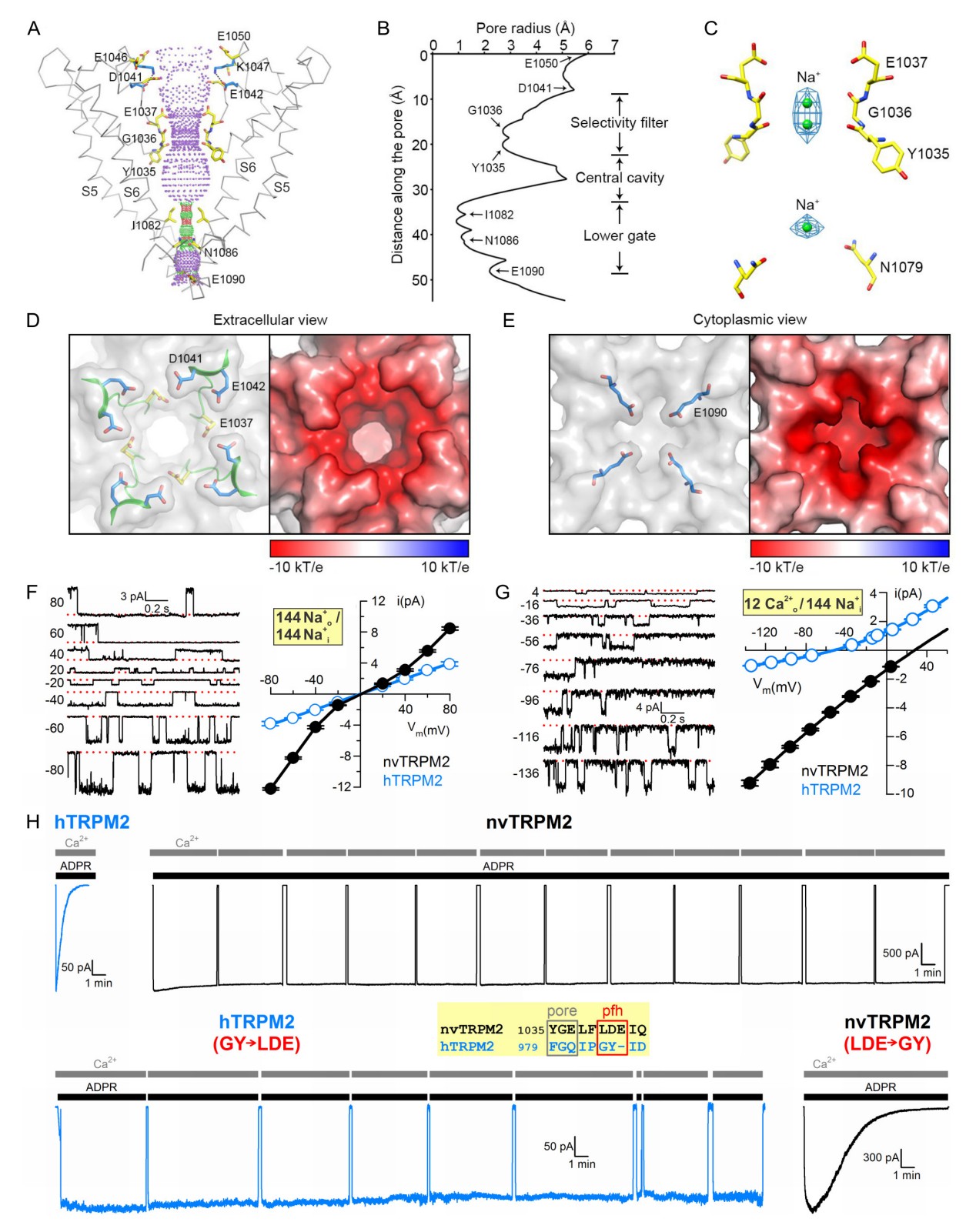

**Figure 3.** Pore of the nvTRPM2 channel. (**A**) Ribbon representation of the ion pore, front and rear subunits are removed for clarity. The dotted mesh distinguishes regions that are too tight (*red*, radius <1.15 Å) or just spatious enough (*green*, 1.15 - 2.30 Å) for a single water molecule to pass, as well as regions with a radius larger than 2.30 Å (*purple*). Residues facing the pore are shown as sticks. The E1042 and K1047 side chains (*blue sticks*) form a salt bridge (*blue dotted line*). (**B**) Van der Waals radius of the pore along the central axis. (**C**) Electron densities in the selectivity filter (*blue meshes*), and

*Figure 3 continued on next page*

*Figure 3 continued*

nearby residues (*sticks*). The Na$^+$ ions within the pore are represented as green spheres. (D) Extracellular view of the pore. Left panel: Surface representation; the selectivity filter and post-filter helix are shown in ribbon, the side chains of some important acidic residues as sticks. Right panel: Electrostatic property of the surface calculated at pH 7 and 0.15 M concentrations of monovalent cations and anions. (E) Cytoplasmic view of the pore. A constricting acidic residue (E1090) is shown as sticks in the left panel. (F–G) Unitary currents (*left*) and unitary current-voltage (i–V) relationships (*right*; *solid black symbols* (mean ± SEM)) of nvTRPM2 channels in symmetrical 144 mM Na$^+$ as the cation (F), or with 12 mM Ca$^{2+}$ in the extracellular (pipette) and 144 mM Na$^+$ in the intracellular (bath) solution (G). *Open blue symbols* (mean ± SEM) in the *i-V* graphs, replotted from (*Tóth and Csanády, 2012*), represent hTRPM2. Smooth fitted curves in (G) were used to estimate reversal potentials under bi-ionic conditions (see Materials and Methods). (H) Inactivation of WT hTRPM2 (*top left*) and of nvTRPM2 with residues 1040–1042 (LDE) replaced by a GY doublet (*bottom right*), and lack of inactivation of WT nvTRPM2 (*top right*) and of hTRPM2 with residues 984–985 (GY) replaced by an LDE triplet (*bottom left*). Membrane potential was −20 mV, currents were activated by cytosolic exposures to 100 µM ADPR (*black bars*; 32 µM for hTRPM2) plus 125 µM Ca$^{2+}$ (*gray bars*). *Inset*: Sequence alignment highlighting target residues (*red box*) swapped between nv and hTRPM2; pfh, post-filter helix. See also *Figure 3—figure supplements 1–3*.
DOI: https://doi.org/10.7554/eLife.36409.009

The following figure supplements are available for figure 3:

**Figure supplement 1.** Structural comparisons between the ion channel pore of nvTRPM2 and of other TRPM family channels.
DOI: https://doi.org/10.7554/eLife.36409.010

**Figure supplement 2.** Disruption of post-filter salt bridge does not cause inactivation.
DOI: https://doi.org/10.7554/eLife.36409.011

**Figure supplement 3.** Alpha-pi-alpha helical transition in S6 of TRP family channels.
DOI: https://doi.org/10.7554/eLife.36409.012

one or two negative charges into the external vestibule of hTRPM2 increased its conductance to ~72 pS, without much effecting relative Ca$^{2+}$ permeability (*Tóth and Csanády, 2012*). Of note, inward unitary current of nvTRPM2 in the presence of 12 mM external Ca$^{2+}$ as the only permeating cation approached −10 pA at negative voltages (*Figure 3G*, *black symbols*), which exceeds the throughput rate expected for a diffusion-limited channel (*Hille, 1992*). This is a clear indication that, due to its negative surface, the local concentration of Ca$^{2+}$ ions in the outer vestibule is far higher than that in the bulk solution. Interestingly, an outermost filter binding site formed by a ring of four carboxylates, and additional negative 'recruitment sites' in the outer vestibule, are also present in the highly Ca$^{2+}$-selective TRPV6 channel (*Saotome et al., 2016*).

Loss of the LDE motif was suggested to be responsible for the fast inactivation of hTRPM2 (*Figure 3H*, *top*, *left*), because its reintroduction abolished rundown (*Figure 3H*, *bottom left*, [*Tóth and Csanády, 2012*]). Consistent with that interpretation, nvTRPM2, which contains this motif, does not inactivate over the time course of an hour (*Figure 3H*, *top*, *right*), but replacing the native LDE triplet with a 'human-like' GY doublet results in clear rundown (time constant 2–3 min; *Figure 3H*, *bottom*, *right*). Thus, the presence and proper positioning of D1041 and E1042 seems important for stabilization of the nvTRPM2 pore, but the E1042-K1047 salt bridge is not essential, because its disruption by mutations K1047A or K1047E does not result in rundown (*Figure 3—figure supplement 2*). Possibly, inactivation is prevented by charge repulsion between the LDE motifs, as suggested (*Tóth and Csanády, 2012*). Because inactivation is not an intrinsic property of nvTRPM2, its transient activation in intact cells (*Kühn et al., 2015*) must reflect some secondary effect caused by cytosolic Ca$^{2+}$ accumulation, such as PIP$_2$ hydrolysis by phospholipase C and/or pore block (cf., *Figure 4E*, below).

As observed in other TRPM family channels, the gate of nvTRPM2 is formed by the S6 bundle crossing which constricts the pore to a diameter of less than 2 Å between residues I1082 and N1086 (*Figure 3A–B*). Because that diameter is too small to allow passage of a water molecule, the observed nvTRPM2 conformation must correspond to a closed gate, as expected in the absence of ADPR. Additionally, S6 shows an α- to π-helix transition in the middle (residues 1073–1077). In TRPV6, for which both open- and closed-state structures have been solved (*McGoldrick et al., 2018*), S6 is α-helical in the closed state, but in the open state the short π helical break induces a kink and a rotation of the S6 inner segment around a 'gating hinge' alanine (A566), which opens the inner gate. Interestingly, the π-helical break in S6 is a common feature seen in most reported TRP channel structures, except for TRPV2 and TRPV6 in the closed state (*Figure 3—figure supplement 3*), and even the gating-hinge alanine is conserved in some TRPM family members such as TRPM1, TRPM3, and TRPM4 (*Guo et al., 2017*). However, although the solved structures of TRPM4 are

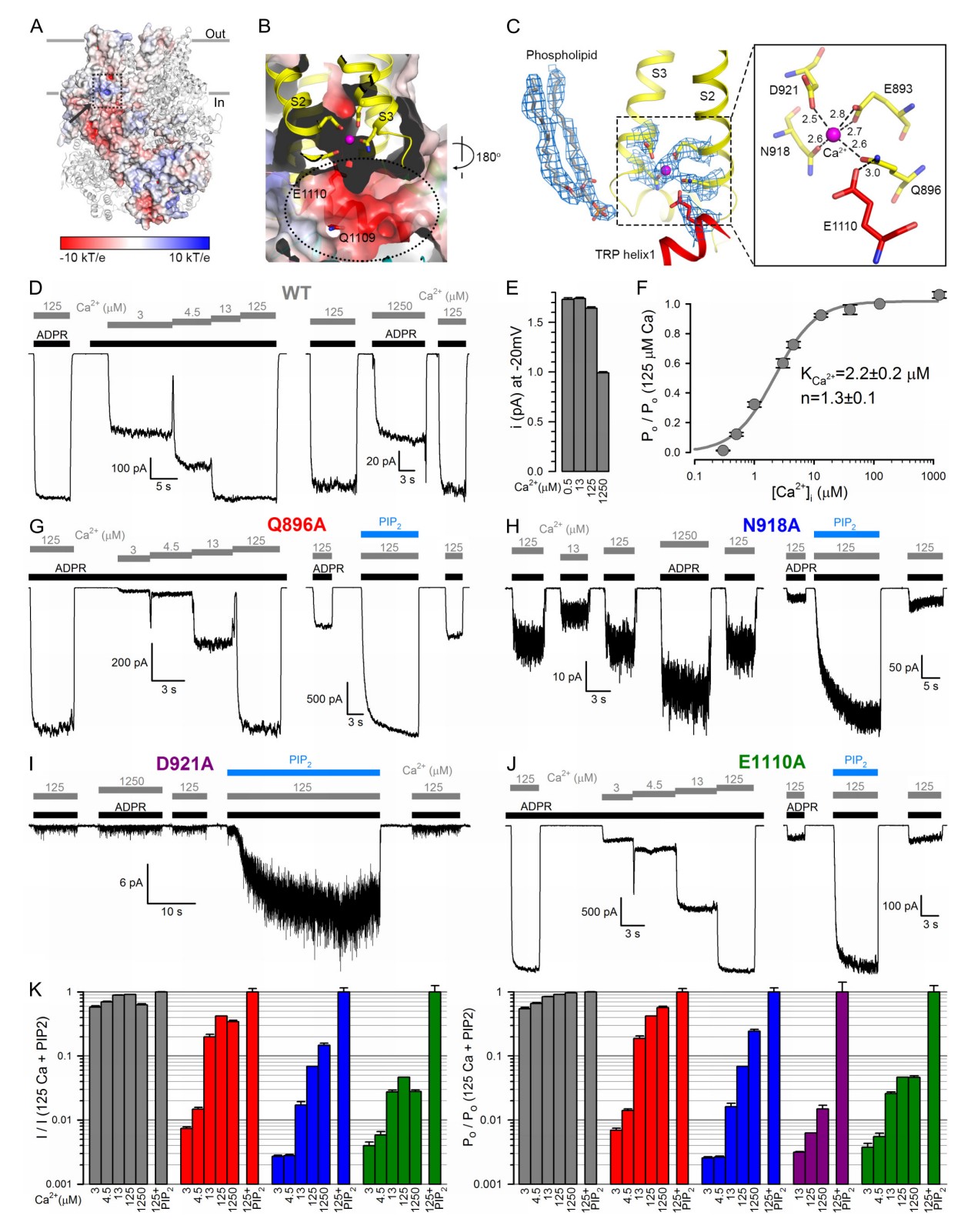

**Figure 4.** The nvTRPM2 Ca$^{2+}$binding site. (**A**) The Ca$^{2+}$ binding site is located close to the inner leaflet of the membrane. One subunit is represented as electrostatic surface, the remaining three are shown as ribbon. The position of the Ca$^{2+}$ binding site (hidden behind surface) is indicated with a dashed box and the vestibule of the peripheral tunnel is marked with an arrow. (**B**) A cross section of the Ca$^{2+}$ binding site. Ca$^{2+}$ is shown as a magenta sphere. Side chains of residues from S2 and S3 involved in Ca$^{2+}$ coordination, as well as nearby side chains of residues in TRP helix 1, are shown as sticks. The

*Figure 4 continued on next page*

*Figure 4 continued*

peripheral tunnel located between the transmembrane and the TRP domain is indicated with a dashed ellipse. (C) Local EM densities and geometry of the $Ca^{2+}$ binding site. The nearby phospholipid with a poorly resolved head group was modeled as a phosphatidic acid. (D, G–J) Macroscopic inward $Na^+$ currents through WT (D), Q896A (G), N918A (H), D921A (I), and E1110A (J) nvTRPM2, evoked by cytosolic exposures to 100 µM ADPR (*black bars*) and various concentrations (in µM) of free $Ca^{2+}$ (*gray bars*), with or without 25 µM dioctanoyl-$PIP_2$ (*blue bars*). Extracellular (pipette) $[Ca^{2+}]$ was ~1 nM, membrane potential was −20 mV. (E) Absolute values of WT nvTRPM2 unitary current amplitudes (mean ± SEM) under conditions similar to those in panel D: at −20 mV membrane potential in the presence of symmetrical 144 mM $Na^+$, but various cytosolic $[Ca^{2+}]$ (in µM). (F) Dependence on cytosolic $[Ca^{2+}]$ of nvTRPM2 open probability ($P_o$; mean ±SEM), normalized to that in 125 µM $Ca^{2+}$ ($P_{o;125}$), calculated as $P_o/P_{o;125}=(I/I_{125})/(i/i_{125})$ ($I$, macroscopic current; $i$, unitary current). *Gray curve* is a fit to the Hill equation with parameters plotted. (K) Dependence on cytosolic $[Ca^{2+}]$ of macroscopic current (*left*; mean ± SEM) and of channel open probability (*right*; mean ± SEM), normalized to the values observed in the presence of 125 µM $Ca^{2+}$ + 25 µM dioctanoyl-$PIP_2$ (see Materials and Methods), for WT (*gray*), Q896A (*red*), N918A (*blue*), D921A (*purple*), and E1110A (*green*) nvTRPM2. Fractional $P_o$ was calculated as in (F), except for D921A for which it was estimated using dwell-time analysis (see Materials and methods) as currents in the absence of $PIP_2$ were too small for reliable cursor measurement. See also *Figure 4—figure supplement 1*.
DOI: https://doi.org/10.7554/eLife.36409.013

The following figure supplement is available for figure 4:

**Figure supplement 1.** Density maps around the $Ca^{2+}$binding site.
DOI: https://doi.org/10.7554/eLife.36409.014

closed, the corresponding region of S6 is π-helical (*Figure 3—figure supplement 3*). Thus, unlike for TRPV6, gate opening in TRPM channels is probably not triggered by the α- to π-helix transition of S6 (*McGoldrick et al., 2018*), but likely resembles opening of TRPV1, in which the S6 bundle crossing opens via direct expansion (*Cao et al., 2013*).

## $Ca^{2+}$ binding site

Although the nvTRPM2 structure was obtained in the presence of trace amounts of $Ca^{2+}$ (free $Ca^{2+}$ concentration in an unbuffered saline with no added $Ca^{2+}$ is in the micromolar range, for example, [*Csanády and Adam-Vizi, 2003*]), clear density for a bound $Ca^{2+}$ ion is seen, occluded into each subunit. The $Ca^{2+}$ binding site, formed by the cytosolic ends of transmembrane helices S2 and S3, and the S2-S3 linker helix, is located at the membrane-cytosol interface (*Figure 4A–C*), and is accessible from the protein surface through a narrow opening ('*peripheral tunnel*'; *Figure 4A*, *arrow*). The bound $Ca^{2+}$ ion is coordinated by the side chains of residues E893 and Q896 of S2 and N918 and D921 of S3 in a pentacovalent geometry, E893 provides two coordinate bonds to the ion (*Figure 4C*, *Figure 4—figure supplement 1*). TRP helix one is located right below the $Ca^{2+}$ site, and the side chain of E1110 (*Figure 4C*) lines the bottom wall of the peripheral tunnel (*Figure 4B*), facilitating access of the cation to its binding site. It also forms a hydrogen bond with the side chain of Q896, keeping the latter properly oriented for $Ca^{2+}$ coordination. These five residues are conserved between TRPM2, 4, 5 and 8 (*Figure 2—figure supplement 3*, *orange boxes*), and the geometry of $Ca^{2+}$ coordination in nvTRPM2 resembles that seen in the $Ca^{2+}$ bound structure of hTRPM4 (*Autzen et al., 2018*).

Compared to hTRPM2, macroscopic nvTRPM2 channel currents show a somewhat higher apparent affinity for activation by cytosolic $Ca^{2+}$ (*Figure 4D*, *left*); a reduction of inward macroscopic current by millimolar cytosolic $Ca^{2+}$ (*Figure 4D*, *right*) reflects a reduction in unitary current amplitude due to pore block (*Figure 4E*). Correcting for the latter effect reveals the $Ca^{2+}$ dependence of channel open probability (*Figure 4F*), characterized by a $K_{1/2}$ of ~2 µM (*Figure 4F*). Truncating any of the five conserved side chains involved in $Ca^{2+}$ coordination (*Figure 4C*) profoundly affected channel activity. The Q896A, N918A, and D921A nvTRPM2 mutants generated functional channels, but in each case current activation required larger free cytosolic $Ca^{2+}$ concentrations (*Figure 4G–I*; *Figure 4K*, *left*), reporting a dramatic loss of $Ca^{2+}$ sensitivity. Correcting for the dose-dependent reduction in unitary current amplitude (*Figure 4E*) allowed calculation of the $Ca^{2+}$ dependence of open probabilities (*Figure 4K*, *right*). Estimated $K_{1/2}$ for channel activation was ~40 µM for Q896A, whereas for N918A and D921A open probability failed to saturate even at 1250 µM ($K_{1/2}$ >1 mM). The E1110A mutant displayed a more modest reduction in apparent $Ca^{2+}$ affinity (*Figure 4J*, *Figure 4K*, *green bars*; $K_{1/2}$ ~12 µM), consistent with the less direct involvement of the E1110 side chain in $Ca^{2+}$ binding (*Figure 4C*). In contrast, E893A channels failed to open even in the presence

of millimolar cytosolic $Ca^{2+}$, underscoring the exquisite importance of this residue in $Ca^{2+}$ coordination (*Figure 4C*).

Interestingly, the $Ca^{2+}$ binding site mutants were all robustly stimulated by $PIP_2$: in contrast to wild-type (WT) channel currents that were stimulated only by 10–20% (*Figure 1B*), the currents for the mutants were enhanced by up to 20-fold upon $PIP_2$ addition (*Figure 4G–J*; *Figure 4K, rightmost bars*). Conversely, suppression of hTRPM2 activity upon $PIP_2$ depletion was shown to be counteracted by raising cytosolic $Ca^{2+}$ (*Tóth and Csanády, 2012*). Clearly, when the side chains involved in its coordination are intact, $Ca^{2+}$ binds so tightly that the binding sites are saturated at 125 µM free $Ca^{2+}$: under such conditions channel open probability is close-to-maximal even when the $PIP_2$ sites are not fully occupied, and adding exogenous $PIP_2$ causes little further current enhancement. However, when a $Ca^{2+}$ coordinating side chain is truncated, the affinity for the ion declines such that its binding sites are scarcely occupied even at 125 µM free $Ca^{2+}$: under such conditions open probability is very low and filling up the $PIP_2$ sites can cause severalfold current stimulation (*Figure 4K*, right). (The large current noise for N918A (*Figure 4H*) and D921A (*Figure 4I*) suggests that for these two mutants open probability is far lower than unity even in the presence of $PIP_2$.) One possible explanation for these findings is that $Ca^{2+}$ and $PIP_2$ stabilize the open state independently of each other. Alternatively, the $PIP_2$ headgroup itself might be directly involved in $Ca^{2+}$ binding. Indeed, density for a phospholipid molecule is seen near the bound $Ca^{2+}$ ion, in the immediate vicinity of the TRP box region of TRP helix 1 (*Figure 4C* and *Figure 2—figure supplement 2A*). The phospholipid headgroup was not clearly resolved, and the phosphate moiety of the modeled phosphatidic acid is too far (~9 Å) from the cation to be involved in its coordination. However, the terminal phosphates of a $PIP_2$ molecule bound in the same position could potentially contact the bound $Ca^{2+}$ ion.

## Cytoplasmic cavities and tunnels and access to the $Ca^{2+}$ binding site

The cytosolic domains form a porous structure which houses a complex system of cavities crucial for channel function. The two ring-like structures formed in the bottom tier by the NTDs and ARDs and in the middle tier by the LHDs surround a common cytoplasmic cavity that extends in a direction perpendicular to the membrane surface (*Figure 5A*). The four stretcher helices form a cross which acts as a joist to subdivide this cavity into two chambers. The lower chamber, contained in the bottom tier, is open at the bottom towards the cytosol, and is mostly occupied by the vertical CC, which fills it like a piston (*Figure 5A, top*). The upper chamber, contained in the middle tier, is bounded from the top by the four TRP helices which act as ceiling joists that run parallel to the stretcher helices and converge into the cytoplasmic mouth of the pore (*Figure 5A, top* and *bottom*). The empty upper chamber represents a relatively secluded volume of ~50000 $A^3$, buried inside the protein (*Figure 5B, Video 1*). At its sides, the upper chamber communicates with the cytosol through eight large lateral windows, four between the bottom and middle tier (NTD and LHD) (*Figure 2F*), and four between the middle and the top tier (LHD and PD) (*Figure 2F–G*). In addition, the upper chamber is directly connected to the $Ca^{2+}$ binding sites through four tunnels (*Figure 5B, 'central tunnels'*). Indeed, the $Ca^{2+}$ binding site can be viewed as a recess in a bent tunnel with a diameters > 4 Å (*Figure 5B, inset*) that connects the upper chamber with the cytosol (*Figure 5C, Video 1*). In nvTRPM2 the surface electrostatics of both the peripheral and the central tunnel is negative (*Figure 5C, left*). Thus, in TRPM2, which is highly $Ca^{2+}$ permeable, $Ca^{2+}$ entering through the open pore can directly access the activating sites, without leaving the protein interior (*Figure 5C, left, yellow arrows*). In contrast, in monovalent-selective TRPM4 the central tunnel likely does not play such a role, as $Ca^{2+}$ ions necessarily have to approach the channel from the cytosol. Interestingly, although both tunnels exist in TRPM4, their surfaces are positive (*Figure 5C, right*).

For hTRPM2 the activating $Ca^{2+}$ sites were shown to be inaccessible to extracellular $Ca^{2+}$ in the closed-pore state, but saturated by extracellular $Ca^{2+}$ entering through the pore in the open-channel state, leading to the suggestion that these sites are intracellular, but buried in some deep crevice, shielded from the protein surface, near the intracellular mouth of the pore (*Csanády and Töröcsik, 2009*). The location of the $Ca^{2+}$ binding sites in nvTRPM2 (*Figure 5B–C*) is consistent with that prediction. Indeed, in the absence of extracellular (pipette) $Ca^{2+}$ (*Figure 5D*), addition and removal of cytosolic $Ca^{2+}$ (*gray bars*) in the continuous presence of ADPR (*black bar*) caused nvTRPM2 currents to activate and deactivate, respectively, regardless of the applied membrane potential (*colored bar* and *shading*); the time constants of current decay in response to cytosolic $Ca^{2+}$ removal (*colored fit lines* and *numbers* (in ms)) attested to a modest intrinsic voltage dependence of gating (*Figure 5F*,

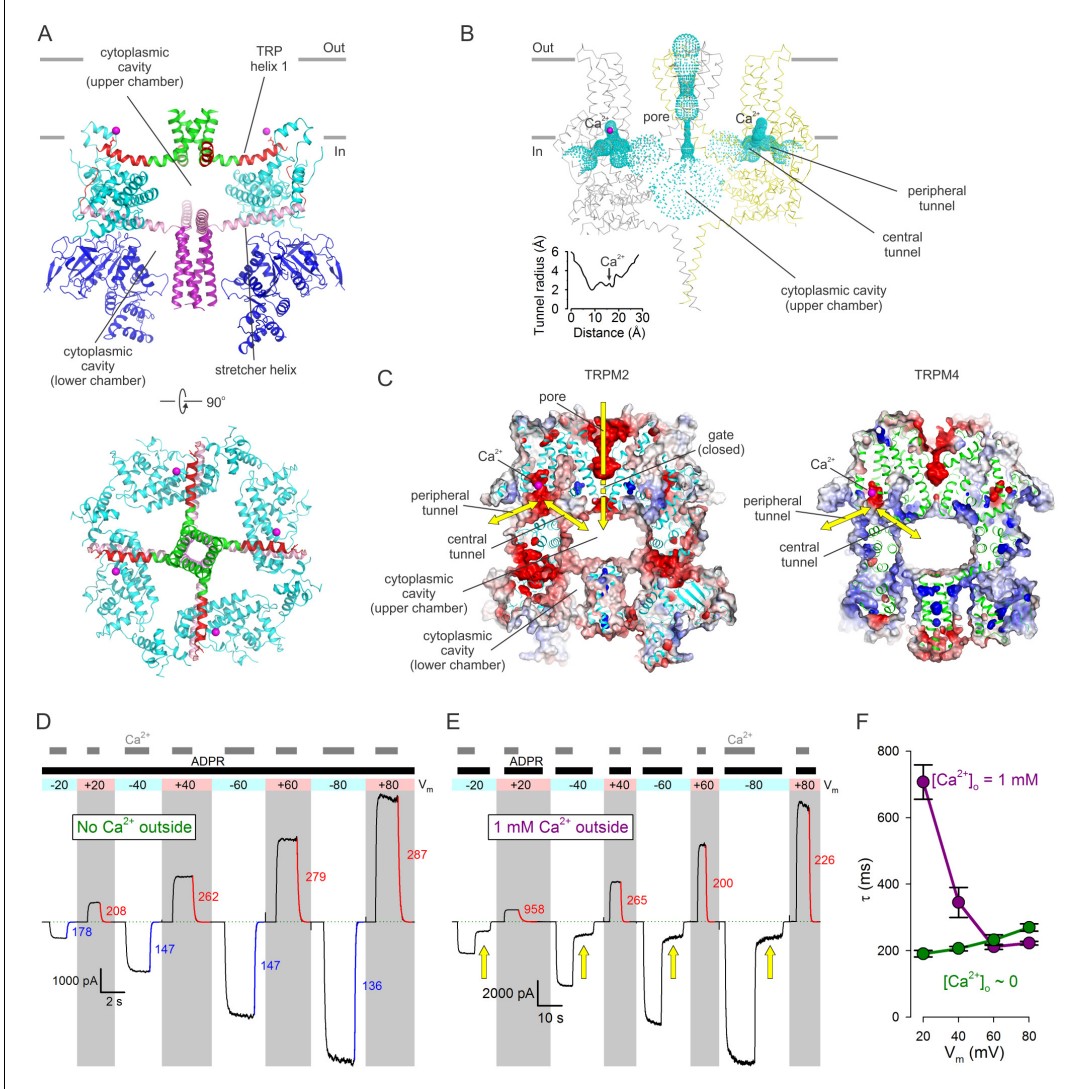

**Figure 5.** Cytoplasmic cavities and tunnels. (**A**) Architecture of the cytoplasmic cavity viewed from an angle parallel (*top*) or perpendicular (*bottom*) to the membrane plane. The NTD (*blue*), the LHD (*cyan*), the cytoplasmic ends of transmembrane helix S6 (*green*), TRP helix 1 and the TRP loop (*red*), the stretcher helix (*pink*), and the CC (*purple*) are shown as cartoon, $Ca^{2+}$ ions as *magenta spheres*, and the E1110 side chain as sticks. In the *top panel* two diagonally opposing subunits are shown, in the *bottom panel* the NTD is removed for clarity. (**B**) Ribbon representation of the top and middle tiers and the CC helices, front and rear subunits omitted for clarity. Dotted mesh represents the contiguous surface that lines the pore, the upper chamber, the central and peripheral tunnels, and the $Ca^{2+}$ binding sites. Inset plots the van der Waals radius of the tunnel along its central axis. (**C**) Longitudinal cross sections through nvTRPM2 (*left*) and hTRPM4 (*right*; PDBID:6BQV), showing connectivities and surface electrostatics for the upper chamber, a central and peripheral tunnel, and the corresponding $Ca^{2+}$ binding site. *Yellow arrows* highlight possible pathways for $Ca^{2+}$ flux. (**D–E**) Macroscopic nvTRPM2 currents evoked at various membrane potentials (*colored bars* and *shading*) by cytosolic exposures to 100 μM ADPR +125 μM $Ca^{2+}$, in the presence of either ~1 nM (**D**) or 1 mM (**E**) free $Ca^{2+}$ in the extracellular (pipette) solution. *Colored curves* are single exponentials fitted to the current decay time courses that follow cytosolic $Ca^{2+}$ removal, *colored numbers* are time constants (in ms). *Green dotted line* marks zero-current level. *Yellow arrows* in (**E**) highlight current fractions that survive removal of cytosolic $Ca^{2+}$. (**F**) Voltage dependence of closing time constants (mean ± SEM) upon cytosolic $Ca^{2+}$ removal, in the presence (*purple symbols*) or absence (*green symbols*) of extracellular $Ca^{2+}$.

DOI: https://doi.org/10.7554/eLife.36409.015

*green symbols*). However, when extracellular $[Ca^{2+}]$ was 1 mM (*Figure 5E*), currents evoked by addition of ADPR+$Ca^{2+}$ (*black and gray bars*) could be completely abolished by cytosolic $Ca^{2+}$ removal only when the membrane potential was positive (*Figure 5E*, *gray shading*), i.e., when the driving force for $Ca^{2+}$ influx through the pore was small. Even at moderately positive voltages (+20 to+40 mV) the decay time constants upon cytosolic $Ca^{2+}$ removal (*red fit lines* and *numbers* (in ms)) were

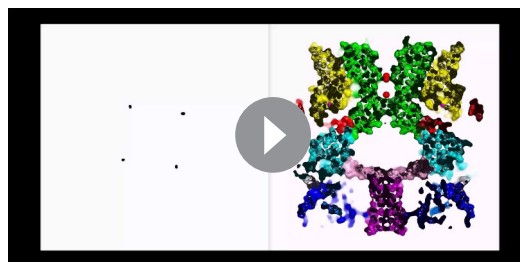

**Video 1.** Cytosolic chambers and tunnels in nvTRPM2. (*Right*) Vertical cross section along nvTRPM2 pore axis: 4 Å slab represented in spacefill with surface rendering. Domain color coding as in *Figure 1E*. Sodium ions in the pore and bound $Ca^{2+}$ ions are shown as *red* and *magenta spheres*, respectively. (*Left*) Sequential horizontal cross sections perpendicular to the nvTRPM2 pore axis: 4 Å slabs represented in spacefill with surface rendering, moving through the structure in 1 Å steps. The actual position of the slab on the left is illustrated by the moving *black box* on the right. Apertures that connect the upper chamber with the

greatly prolonged by the presence of external $Ca^{2+}$ (*Figure 5F*, *purple symbols*), approaching those measured in its absence (*Figure 5F*, *green symbols*) only at extremely positive voltages (+60 to +80 mV) at which the driving force for $Ca^{2+}$ influx is negligible. Such slowing of channel closure by external $Ca^{2+}$ has been described for hTRPM2 (*Csanády and Törőcsik, 2009*), and suggests that, as long as the gate is open, $Ca^{2+}$ entering through the nvTRPM2 pore can directly reach the $Ca^{2+}$ sites through the central tunnels, and keep them saturated even while the protein surface is continuously rinsed with a $Ca^{2+}$-free solution. However, at positive voltages, and with the cytosolic face exposed to continuous $Ca^{2+}$-free wash, the $Ca^{2+}$ sites and the cytoplasmic cavity are depleted of $Ca^{2+}$ as soon as the gate closes, too rapidly to allow it to reopen.

By contrast, at negative membrane potentials (*Figure 5E*, *white shading*, *Figure 6B*) that provide a large driving force for the influx of millimolar extracellular $Ca^{2+}$, a substantial fraction of

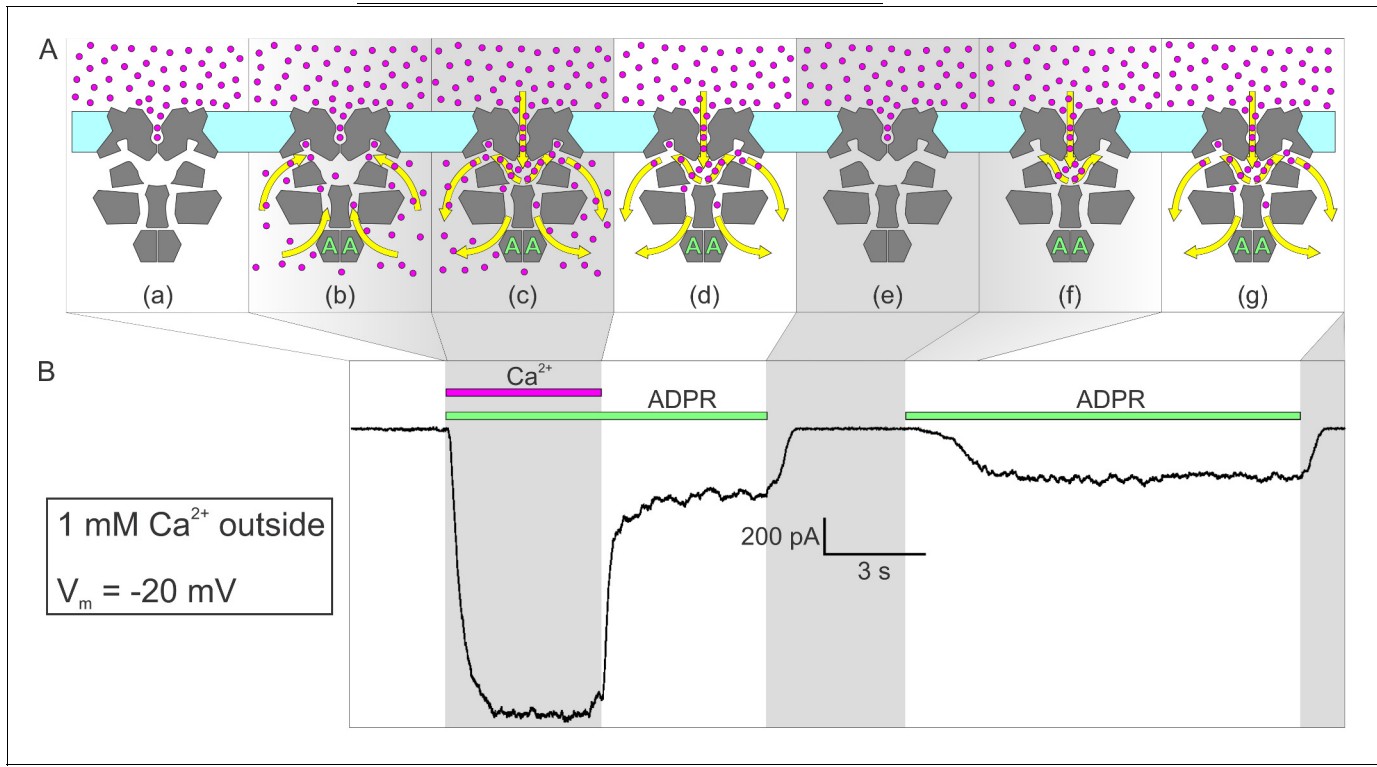

**Figure 6.** Activation by ADPR of nvTRPM2 current in the presence of external but absence of cytosolic $Ca^{2+}$. (**A**) Cartoon interpretation of the molecular events that occur during consecutive time intervals (*sections a-g*, also identified by intermittent gray shading in (**A**)-(**B**)) of the current recording in (**B**). Membrane, *light cyan*; nvTRPM2 protein, *dark gray*; $Ca^{2+}$ ions, *magenta spheres*; ADPR, *green letters 'A'*; direction of $Ca^{2+}$ flow, *yellow arrows*. (**B**) In the presence of 1 mM external $Ca^{2+}$, at −20 mV membrane potential, a fraction of the macroscopic nvTRPM2 current evoked by cytosolic exposure to 100 μM ADPR +125 μM free $Ca^{2+}$ survives cytosolic $Ca^{2+}$ removal, and subsides only upon removal of ADPR (see *Figure 5E*). A second application of 100 μM ADPR, without cytosolic $Ca^{2+}$, activates a current comparable to that which survived prior $Ca^{2+}$ removal. Note delayed current activation following exposure to ADPR alone.
DOI: https://doi.org/10.7554/eLife.36409.017

nvTRPM2 current survived cytosolic $Ca^{2+}$ removal (*Figure 5E*, *yellow arrows*), and subsided only when ADPR was also removed. One possible explanation is that at negative transmembrane voltages the number of $Ca^{2+}$ ions accumulated in the cytoplasmic cavity during an open-pore event is too large to be completely dissipated during a subsequent closed event: the pore reopens before all $Ca^{2+}$ ions would have left the cavity. This results in a low but substantial steady-state channel activity (open probability ~0.15–0.3; *Figure 5E*, *white shading* and *yellow arrows*; *Figure 6A–B*, *section (d)*) during which $[Ca^{2+}]$ in the upper chamber of each channel fluctuates between a very high (during open events) and a low but non-zero value (during closed events). Thus, whereas all four $Ca^{2+}$ sites of a channel likely become saturated during each open event, some will lose $Ca^{2+}$ during the subsequent closed event. Because channel opening rate is a property of the closed channel, the surviving current (*Figure 5E*, *yellow arrows*) reflects channel opening rate under sub-saturating conditions, explaining the reduction in open probability upon removal of bulk cytosolic $Ca^{2+}$.

Alternatively, given the large density of channels in the patch (up to ~2000 channels/$\mu m^2$; corresponding to a pore-to-pore distance of ~20 nm), the current that survives cytosolic $Ca^{2+}$ removal might reflect channel cross-talk, that is, $Ca^{2+}$ entering through a still-open pore re-opening a nearby closed channel, despite continuous rinsing of the cytosolic surface. Indeed, low millimolar concentrations of calcium buffer (especially of slow buffers like EGTA) were predicted to be inefficient for buffering cytosolic $Ca^{2+}$ around a channel pore (*Stern, 1992*). Although we cannot strictly exclude the possibility that some of the $Ca^{2+}$ entering through open nvTRPM2 pores does accumulate at the internal surface of our patches, several arguments suggest that this is unlikely to happen. First, based on the unitary conductances measured with $Na^+$ or $Ca^{2+}$ as the charge carrier (*Figure 3F–G*), when the extracellular solution contains 140 mM $Na^+$ and 1 mM $Ca^{2+}$ (*Figure 5E*, *Figure 6*), the majority (~74%) of inward nvTRPM2 currents are carried by $Na^+$ ions (which account for ~85% of entering cations). Second, in addition to EGTA, our cytosolic solution contains 140 mM gluconate. Although gluconate is a low-affinity buffer, at 140 mM it binds ~88% of total cytosolic $Ca^{2+}$ (*Csanády and Töröcsik, 2009*), and its high concentration precludes its saturation. Third, Stern's calculation applies to a static environment (like the cytosol of an intact cell during whole-cell recording) in which buffering around the pore is limited partly by the kinetics of $Ca^{2+}$ binding to the buffer, and partly by the rate of diffusion of the buffer ligand, because the buffer molecules in the vicinity of the pore become rapidly saturated and must be replaced by unbound buffer molecules via diffusion. Here the cytosolic face of the patch is immersed into a rapidly flowing solution. Considering the flow rate of ~1 cm/s of our bath solution and the ~10 nm diameter of a channel protein (*Figure 2C*), the entire solution volume in contact with the surface of a channel is displaced by fresh $Ca^{2+}$-free solution every ~1 $\mu s$, during which time ~2 $Ca^{2+}$ ions enter through its pore (assuming a $Ca^{2+}$ current of 0.6 pA per open channel). Thus, the vectorial flushing effect is likely more important here than diffusion or the kinetics of $Ca^{2+}$ binding to buffers – unless the shape of our patches is concave towards the bulk solution to an extent that severely limits access of the flowing bath solution to the patch surface. Fourth, the perhaps most compelling argument against $Ca^{2+}$ accumulation at the patch surface being responsible for the residual currents in *Figure 5E*, (*yellow arrows*) is our finding that at negative voltages, in the presence of external $Ca^{2+}$, nvTRPM2 currents activate, albeit with a delay, upon addition of cytosolic ADPR even in the absence of cytosolic $Ca^{2+}$ (*Figure 6B*, *right*): the magnitude of that 'spontaneous' current is identical to that which survives removal of pre-applied cytosolic $Ca^{2+}$ (*Figure 6B*, *left*). Because the spontaneous current activates from a state in which all pores in the patch have been closed for several seconds (*Figure 6A–B*, *section (e)*), a duration more than sufficient to completely wash off even very high concentrations of pre-applied cytosolic $Ca^{2+}$ (cf., *Figure 1A*), it cannot be explained by bulk $Ca^{2+}$ accumulation. Rather, its delayed activation must reflect a very low 'spontaneous' channel opening rate of ADPR-bound channels in the absence of bound $Ca^{2+}$ (*Figure 6A*, *section (f)*): upon the first such spontaneous opening of a channel its upper chamber is immediately flooded by $Ca^{2+}$ which henceforth maintains that channel in an active state (*Figure 6A*, *section (g)*).

In conclusion, we have presented here the cryo-EM structure of TRPM2 from *Nematostella vectensis*, which is functionally similar to human TRPM2, in complex with bound $Ca^{2+}$. The overall structure of nvTRPM2 resembles that of other TRPM family channels, but important differences in local geometry and surface polarity explain many of its unique functional features. In particular, the larger pore diameter and larger negative surface charge of both pore vestibules in nvTRPM2 explains its higher $Ca^{2+}$ permeability and larger conductance compared to TRPM4. The external vestibule of

nvTRPM2 is stabilized by a negatively charged short helix. Loss of the corresponding sequence segment explains inactivation of human TRPM2 (*Tóth and Csanády, 2012*). Changes in $Ca^{2+}$- and $PIP_2$-sensitivity upon truncation of $Ca^{2+}$-coordinating amino acid side chains suggest a mechanism for gating regulation by these two channel cofactors. The secluded location of the activating $Ca^{2+}$ sites in the protein interior in a recess of a narrow tunnel that connects the upper chamber of the cytoplasmic cavity to the cytosol, and the negative surface charge of this tunnel in TRPM2, explain the unique dependence of TRPM2 activity on intra- and extracellular $Ca^{2+}$. Thus, although the $Ca^{2+}$ sites are intracellular in both TRPM4 and TRPM2, their regulation through $Ca^{2+}$ is fundamentally different. TRPM4 activity requires a continuous supply of cytosolic $Ca^{2+}$ that enters through its peripheral tunnel. In contrast, ADPR-bound TRPM2 requires only an initiator $Ca^{2+}$ spark from the cytosol (*Figure 6A*, section (b)), to then remain activated by $Ca^{2+}$ ions that enter through its pore and access the binding site through the central tunnel (*Figure 6A*, section (c)). Indeed, extremely $Ca^{2+}$ permeable nvTRPM2 does not even require an initiator spark: infrequent spontaneous openings of ADPR-bound channels allow for self-activation using exclusively extracellular $Ca^{2+}$ (*Figure 6A*, sections (f-g)) – that mechanism might be relevant even for moderately $Ca^{2+}$ permeable hTRPM2 under the static conditions of a living cell. Because the ADPR-binding NUDT9H domain was not resolved here, further studies will be required to address the structural underpinnings of TRPM2 regulation by ADPR.

## Materials and methods

### Key resources table

| Reagent type (species) or resource | Designation | Source or reference | Identifiers | Additional information |
|---|---|---|---|---|
| cell line (*Spodoptera frugiperda*) | Sf9 | ATCC | CRL-1711 | |
| cell line (*Homo sapiens*) | HEK293S GnTI- | ATCC | CRL-3022 | |
| biological sample (*Xenopus laevis*) | Xenopus laevis oocytes | African Reptile Park < mandyvorster@xsinet.co.za> | RRID:NXR_0.0080 | |
| commercial assay or kit | CNBR-activated sepharose beads | GE Healthcare | 17043001 | |
| commercial assay or kit | Superose 6, 10/300 GL | GE Healthcare | 17517201 | |
| commercial assay or kit | HiSpeed Plasmid Midi Kit | Qiagen | Catalog #12643 | |
| commercial assay or kit | QuikChange XL Site-Directed Mutagenesis Kit | Agilent Technologies | Catalog #200521 | |
| commercial assay or kit | mMESSAGE mMACHINE T7 Transcription Kit | ThermoFisher | Catalog #AM1344 | |
| chemical compound, drug | 2,2-didecylpropane-1,3-bis-β-D-maltopyranoside (LMNG) | Anatrace | NG310 | |
| chemical compound, drug | Cholesteryl hemisuccinate (CHS) | Anatrace | CH210 | |
| chemical compound, drug | Digitonin | Sigma-Aldrich | D141 | |
| chemical compound, drug | sf-900 II SFM medium | Gibco | Cat#10902088 | |
| chemical compound, drug | Cellfectin II reagents | Invitrogen | Cat#10362100 | |
| chemical compound, drug | Freestyle 293 medium | Gibco | Cat#12338018 | |
| chemical compound, drug | HI FBS | Gibco | Cat#16140071 | |
| chemical compound, drug | Antibiotic-Antimycotic (100X) | Gibco | Cat#15240062 | |

*Continued on next page*

*Continued*

| Reagent type (species) or resource | Designation | Source or reference | Identifiers | Additional information |
|---|---|---|---|---|
| chemical compound, drug | Gentamicin sulphate | Sigma-Aldrich | G1397-10mL | |
| chemical compound, drug | Collegenase type II | Gibco by life technologies | 17107–0125 | |
| chemical compound, drug | Adenosine 5′-diphosphoribose sodium salt | Sigma-Aldrich | A0752 | |
| chemical compound, drug | PtdIns-(4,5)-P2 (1,2-dioctanoyl) (sodium salt) | Cayman Chemical | 64910 | |
| software, algorithm | Seriel EM | DOI: 10.1016/j.jsb.2005.07.007 | http://bio3d.colorado.edu/SerialEM | |
| software, algorithm | MotionCor2 | DOI: 10.1038/nmeth.4193 | http://msg.ucsf.edu/em/software/motioncor2.html | |
| software, algorithm | Gctf | DOI: 10.1016/j.jsb.2015.11.003 | https://www.mrc-lmb.cam.ac.uk/kzhang/ | |
| software, algorithm | Gautomatch | other | https://www.mrc-lmb.cam.ac.uk/kzhang/ | Downloaded from a personal URL |
| software, algorithm | RELION 2.1 | DOI: 10.7554/eLife.18722 | http://www2.mrc-lmb.cam.ac.uk/relion | |
| software, algorithm | SWISS-MODEL | DOI: 10.1093/nar/gku340 | https://swissmodel.expasy.org | |
| software, algorithm | COOT | DOI: 10.1107/S0907444910007493 | https://www2.mrc-lmb.cam.ac.uk/personal/pemsley/coot | |
| software, algorithm | PHENIX | DOI: DOI: 10.1107/S0907444909052925 | https://www.phenix-online.org | |
| software, algorithm | Blocres | DOI: 10.1016/j.jsb.2006.06.006 | https://lsbr.niams.nih.gov/bsoft/programs/blocres.html | |
| software, algorithm | MolProbity | DOI: 10.1107/S0907444909042073; 10.1093/nar/gkm216 | http://molprobity.biochem.duke.edu | |
| software, algorithm | Chimera | DOI: 10.1002/jcc.20084 | https://www.cgl.ucsf.edu/chimera | |
| software, algorithm | Pymol | PyMOL | http://www.pymol.org | |
| software, algorithm | HOLE | PMID: 9195488 | http://www.holeprogram.org | |
| software, algorithm | APBS | DOI: 10.1093/nar/gkm276; 10.1073/pnas.181342398 | http://www.poissonboltzmann.org | |
| software, algorithm | Pclamp9 | Molecular Devices | RRID:SCR_011323 | |
| other | R1.2/1.3 400 mesh Au holey carbon grids | Quantifoil | 1210627 | |

## Cell culture

Insect cells were cultured at 28°C in sf-900 II SFM medium (GIBCO) supplemented with 5% FBS and 1% Antibiotic-Antimycotic. Mammalian cells were grown at 37°C in Freestyle 293 (GIBCO) supplemented with 2% FBS and 1% Antibiotic-Antimycotic. All cells were maintained with 8% $CO_2$ and 80% humidity.

## Protein expression and purification

The *Nematostella vectensis* (nv) *TRPM2* gene was synthesized into the pRML-13 BacMam expression vector (generous gift from Eric Gouaux) with a C-terminal GFP tag attached (General Biosystems). The plasmid was transformed into DH10 Bac cells (Invitrogen, Waltham, MA USA) to produce bacmid DNA, which was transfected into Sf9 cells (ATCC, Catalog#: ATCC CRL-1711) to generate recombinant baculoviruses. 10% (v/v) P3 virus was added to HEK 293S GnTI⁻ cells (ATCC, Catalog#: ATCC CRL-3022) at $3 \times 10^6$ cells/ml. After 12 hr incubation at 37°C, protein expression was induced

by 10 mM sodium butyrate at 30°C for 48 hr (*Goehring et al., 2014*). Cells were harvested by centrifugation at 4,000 rpm for 20 min.

For protein purification, the cells were resuspended and homogenized in lysis buffer (50 mM Tris-HCl pH 8.0, 2 mM MgCl$_2$, 200 mM NaCl, 20% Glycerol, and 1 mM DTT) supplemented with protease inhibitors (1 mM phenylmethanesulfonyl fluoride (PMSF), 1 mM benzamidine, 1 µg/ml aprotinin, 100 µg/ml trypsin inhibitor, 1 µg/ml leupeptin, and 1 µg/ml pepstatin) and DNase (2 µg/ml). Membranes were solubilized with 1% 2,2-didecylpropane-1,3-bis-β-D-maltopyranoside (LMNG) and 0.1% cholesteryl hemisuccinate (CHS) at 4°C for 2 hr. After centrifugation at 75,000 g for 1 hr, the supernatant was mixed with GFP nanobody-coupled resin at 4°C for 2 hr. The resin was washed with 20 column volumes of Buffer A (20 mM Tris-HCl pH 8.0, 150 mM NaCl, 0.06% digitonin, and 1 mM DTT) to exchange LMNG and CHS with digitonin, and then incubated with PreScission protease (~10:1 w/w protein-to-enzyme ratio) at 4°C overnight to remove the C-terminal GFP tag. The GST-tagged protease was removed by binding to a glutathione-sepharose resin (GE HEalthcare). The concentrated nvTRPM2 protein was further purified by gel filtration in Buffer A on a Superose 6 10/300 column (GE Healthcare).

## EM sample preparation, data collection, and processing

Gel filtration peak fractions in Buffer A were concentrated to 5 mg/ml protein, 3 µl of fresh protein was placed onto Quantifoil R1.2/1.3 400 Au holey carbon grids (Quantifoil) and blotted using Vitrobot (FEI). Humidity was set to 100%, blotting time to 3 s and force to 0. After flash freezing in liquid ethane, the grids were stored in liquid nitrogen until screening and data collection.

The grids were initially screened on a 200 kV Talos Arctica (FEI) microscope, selected grids were then loaded into a 300 kV Titan Krios (FEI) microscope with a K2 summit detector (Gatan). Data were collected in super-resolution mode using Serial EM (*Mastronarde, 2005*). Physical pixel size was 1.03 Å and dose rate was 8 electrons/pixel/second. Images were exposed for 10 s, subdivided into 50 frames, amounting to a total dose of ~75 electrons/Å$^2$. A total of 1619 images were collected. After manual inspection to remove poor quality images, 1550 images were used for further processing.

Beam-induced sample motion was corrected for using MotionCor2 (*Zheng et al., 2017*). Contrast transfer function (CTF) estimation was performed using Gctf (*Zhang, 2016*). Particle auto-picking was done by Gautomatch (http://www.mrc-lmb.cam.ac.uk/kzhang), selected particles were manually inspected to remove false-positives and supplement false-negatives. Finally, a total number of 196,198 particles were input to RELION2 (*Kimanius et al., 2016*) for further classification and refinement. 2D classification sorted out 144,717 good particles which were used for subsequent 3D classification. The initial reference map for 3D classification was generated by ab-initio reconstruction in cryoSPARC (*Punjani et al., 2017*). After 3D classification, the two good classes out of three, containing 104,268 particles, were combined for 3D auto-refine. The refinement, first carried out by using a loose mask which included the entire micelle densities, yielded a 3.22 Å map. Subsequent application of a tighter mask just around the protein density for the final local searches of 3D auto-refine increased the resolution to 3.11 Å. After post-processing, the final resolution was further improved to 3.07 Å using the 0.143 cutoff criterion. The map shown in this paper was sharpened with a B-factor of −101 Å$^2$ and low-pass filtered to 3.07 Å during post-processing.

## Model building and refinement

The transmembrane domain (TMD) was modeled using SWISS-MODEL (*Biasini et al., 2014*) based on the TRPV1 structure (PDB: 3J5P) (*Liao et al., 2013*), roughly fitted into the cryo-EM map in Chimera (*Pettersen et al., 2004*), and then manually adjusted to the density map in Coot (*Emsley et al., 2010*). All other regions of the protein were built de novo in Coot, since the densities for most of the side chains were quite distinct. The N-Acetylglucosamine modifications of residue N1017 were very clear, and further confirmed the validity of model building around this region. Some obvious phospholipid-like densities were seen around the TMD. As these accounted only for parts of phospholipids, and the densities for the head groups were particularly poorly resolved, they were filled with various truncated versions of 1-palmitoyl-2-oleoyl-sn-glycero-3-phosphocholine (POPC) to maximally fit the densities. In addition, a few flatter densities that could not be fitted with phospholipids were filled with cholesterol. The final structure model (tetramer) contained 4244

protein residues (6–23, 41–51, 86–546, 573–755, 812–1136, and 1148–1210), 4 $Ca^{2+}$ ions, 3 $Na^{+}$ ions, 60 lipid molecules, and 8 N-Acetylglucosamine molecules. The EM density for the C-terminal NUDT9H domain (residues 1271–1551) was not observed. Because the very N-terminal segments (6–23 and 41–51) had poor density and were disconnected from the rest of the protein, their registers are somewhat uncertain and they were built as poly-alanines.

The refinement was performed in Phenix (*Adams et al., 2010*) with secondary structure and C4 non-crystallographic symmetry (NCS) restraints. For cross-validation (*Brown et al., 2015*), the two half maps generated by 3D auto-refine were sharpened with the same B-factor and low-pass filter as used for post-processing. The model was randomly displaced by 0.5 Å and then refined against half map1 using Phenix. The Fourier shell correlation (FSC) curves were calculated between the refined model and different maps (full map, half map1, and half map2), respectively. The small differences between the FSC curves derived from half map1 and half map2 indicated that the model was not over-fitted during refinement.

Local resolution estimation was carried out by Blocres (*Heymann and Belnap, 2007*). Validation of geometries was performed using MolProbity (*Chen et al., 2010*; *Davis et al., 2007*). All the structure figures were generated using Pymol (http://www.pymol.org), Chimera, HOLE (*Smart et al., 1996*), and APBS (*Baker et al., 2001*; *Dolinsky et al., 2007*).

## Isolation and maintenance of *Xenopus laevis* oocytes

Ovarian lobes were removed from anaesthetized *Xenopus laevis* [RRID:NXR_0.0080] following a IACUC-approved protocol. Oocytes were defolliculated by treatment with Type II collagenase (GIBCO) and stored at 18°C in a frog Ringer's solution supplemented with 1.8 mM $CaCl_2$ plus 50 µg/ml gentamycin sulfate (Sigma).

## Functional expression of nvTRPM2 in *Xenopus laevis* oocytes

The nv*TRPM2* gene was synthesized into the pGEMHE expression vector (General Biosystems). Mutations were introduced using Stratagene QuikChange. cDNA was transcribed in vitro using T7 polymerase (mMESSAGE mMACHINE T7 kit Thermo Scientific), and cRNA stored at −80°C. *Xenopus laevis* oocytes were injected (Drummond Nanoject) with 0.1–10 ng of WT or mutant nvTRPM2 cRNA, and recordings were done 1–3 days after injection.

## Excised inside-out patch-clamp recording

Macroscopic and unitary nvTRPM2 currents were recorded in excised inside-out patches at 25°C in symmetrical 140 mM Na-gluconate based solutions to avoid activation of endogenous $Ca^{2+}$-activated chloride currents, as described earlier for hTRPM2 (*Csanády and Törőcsik, 2009*). The tip of the patch pipette was filled to ~1 cm height with 140 mM Na-gluconate, 2 mM Mg-gluconate$_2$, 10 mM HEPES (pH = 7.4 with NaOH; free $[Ca^{2+}]$~0.5 µM); 1 mM Na-EGTA or 8 mM Ca(gluconate)$_2$ was added to obtain free $[Ca^{2+}]$ of ~1 nM (*Figures 1*, *4* and *5D*) or ~1 mM (*Figures 5E* and *6*). The pipette electrode was placed into a 140 mM NaCl-based solution carefully layered on top. Bath solution contained 140 mM Na-gluconate, 2 mM Mg-gluconate$_2$, 10 mM HEPES (pH 7.1 with NaOH), and either 1 mM EGTA (to obtain 'zero' (~1 nM) $Ca^{2+}$), or 20 µM to 10 mM Ca-gluconate$_2$ (to obtain 3 to 1250 µM free $[Ca^{2+}]$). For unitary current measurements under biionic conditions (*Figure 3G*) the pipette solution contained 10 mM $CaCl_2$, 0.5 mM $MgCl_2$, and 10 mM HEPES (pH = 7.4 with Ca(OH)$_2$). $Na_2$-ADPR (Sigma), Dioctanoyl-PI(4,5)P$_2$ (Cayman Chemical), and poly-L-lysine (Sigma) were added to the bath solution from 200 mM, 2.5 mM, and 15 mg/ml aqueous stock solutions, respectively. The continuously flowing bath solution was exchanged using computer-driven electronic valves; solution exchange time constant was <100 ms. Macroscopic and unitary nvTRPM2 currents were recorded at the indicated membrane potentials (Axopatch 200B, Molecular Devices), digitized at 10 kHz (Digidata 1440A, Pclamp9 [RRID:SCR_011323], Molecular Devices), and filtered at 2 kHz.

## Analysis of current recordings

For recordings under asymmetrical ionic conditions (*Figure 3G*) liquid junction potential was experimentally determined and corrected off-line. Channel currents were digitally low-pass filtered at 200 Hz before analysis. Unitary current amplitudes (*Figure 3F–G*, *Figure 4E*) were obtained by fitting sums of Gaussian functions to all-points histograms. The unitary current (*i*)-voltage (*V*) relationship

under bionic conditions (*Figure 3G*), was well fitted by the Goldmann-Hodgkin-Katz equation for hTRPM2 (*Figure 3G*, *blue curve*), but not for nvTRPM2. Thus, for nvTRPM2 the *i-V* plot was fitted to a quadratic function to estimate the reversal potential. Because for nvTRPM2 unitary currents could not be resolved at membrane potentials $>+ 4$ mV, the extrapolated reversal potential ($\sim+28$ mV) was also verified in, and found roughly consistent with, macroscopic recordings.

Fractional current activation in 100 µM ADPR by test concentrations of cytosolic $Ca^{2+}$, or by 125 µM $Ca^{2+}$ + 25 µM dioctanoyl-PIP$_2$, were calculated by dividing steady current in the test segment ($I$) with that in 125 µM $Ca^{2+}$ ($I_{125}$) in the same patch. Open probabilities ($P_o$) normalized to that in 125 µM $Ca^{2+}$ ($P_{o;125}$), were calculated as $P_o/P_{o;125}=(I/I_{125})/(i/i_{125})$ ($i$, unitary current; $i_{125}$, unitary current in 125 µM cytosolic $Ca^{2+}$[see *Figure 4E*]). To emphasize several orders-of-magnitude reductions in maximal open probability for the $Ca^{2+}$ site mutants, normalized currents and open probabilities hence obtained are shown in *Figure 4K* rescaled by their values measured in 125 µM $Ca^{2+}$ + 25 µM PIP$_2$, i.e., $I/I_{125+PIP2}=(I/I_{125})/(I_{125+PIP2}/I_{125})$ and $P_o/P_{o;125+PIP2}=(P_o/P_{o;125})/(P_{o;125+PIP2}/P_{o;125})$. For D921A nvTRPM2 channel currents in the absence of PIP$_2$ were too small for reliable cursor measurement. Thus, for this mutant $i$ and $N{\cdot}P_o$ in 13, 125, or 1250 µM cytosolic $Ca^{2+}$ was estimated using dwell-time analysis, and fractional $P_o$ under such conditions (*Figure 4K*, *right*, *purple bars*) calculated as $P_o/P_{o;125+PIP2}=(i{\cdot}N{\cdot}P_o)/(I_{125+PIP2})$.

Macroscopic current relaxations (*Figure 5D–E*) were fitted by single-exponential functions using least-squares. At negative membrane potentials, in the presence of external $Ca^{2+}$, current decay time courses upon cytosolic $Ca^{2+}$ removal (*Figure 5E*) do not reflect channel closing rate, as opening rate remains non-zero under such conditions: thus, these current segments were not fitted.

### Statistics

The quantification and statistical analyses for the structural parts are integral outputs of the software and algorithms used. Electrophysiological data are given as mean ± SEM of measurements from $\geq 5$ (typically ~20) segments of recording, from $\geq 3$ (typically ~10) patches.

### Data and software availability

Cryo-EM density map of nvTRPM2 has been deposited in the electron microscopy data bank (EMDB) under accession code EMD-7542. Atomic coordinates of nvTRPM2 have been deposited in the protein data bank (PDB) under accession code: 6CO7.

## Acknowledgements

We thank Mark Ebrahim and Johanna Sotiris at the Rockefeller Evelyn Gruss Lipper Cryo-Electron Microscopy Resource Center for assistance in data collection. This work was supported by Hungarian Academy of Sciences (HAS) Lendület grant LP2017-14/2017 and an International Early Career Scientist grant from the Howard Hughes Medical Institute to LC. ZZ. is supported by the Charles H Revson fellowship in Biomedical Science. BT. is supported by the ÚNKP-17–4 New National Excellence Program of the Ministry of Human Capacities of Hungary. AS. is a János Bolyai Research Fellow. JC. is a Howard Hughes Medical Institute investigator.

## Additional information

### Competing interests

László Csanády: Reviewing editor, *eLife*. The other authors declare that no competing interests exist.

### Funding

| Funder | Grant reference number | Author |
| --- | --- | --- |
| Howard Hughes Medical Institute | International Early Career Scientist grant | László Csanády |
| Magyar Tudományos Akadémia | Lendület grant LP2017-14/2017 | László Csanády |

| Charles H. Revson Foundation | Charles H. Revson fellowship in Biomedical Science | Zhe Zhang |
| Ministry of Human Capacities of Hungary | ÚNKP-17-4 New National Excellence Program | Balázs Tóth |
| Howard Hughes Medical Institute | HHMI Investigator | Jue Chen |
| Magyar Tudományos Akadémia | János Bolyai Research Fellowship | Andras Szollosi |

The funders had no role in study design, data collection and interpretation, or the decision to submit the work for publication.

## Author contributions
Zhe Zhang, Purified the protein, Determined, refined, and analyzed the structure, Wrote the manuscript; Balázs Tóth, Performed the electrophysiology experiments and analyzed the data; Andras Szollosi, Conceptualized the study, Purified the protein; Jue Chen, Analyzed the structure, Wrote the manuscript; László Csanády, Conceptualized the study, Analyzed the structure, Wrote the manuscript with input from all authors

## Author ORCIDs
Andras Szollosi https://orcid.org/0000-0002-5570-4609
Jue Chen https://orcid.org/0000-0003-2075-4283
László Csanády http://orcid.org/0000-0002-6547-5889

## Ethics
Animal experimentation: This study was performed in strict accordance with the recommendations in the Guide for the Care and Use of Laboratory Animals of the National Institutes of Health. All of the animals were handled according to approved institutional animal care and use committee (IACUC) protocols of Semmelweis University (last approved 06-30-2016, expiration 06-30-2021).

## Decision letter and Author response
Decision letter https://doi.org/10.7554/eLife.36409.024
Author response https://doi.org/10.7554/eLife.36409.025

# Additional files
## Supplementary files
• Transparent reporting form
DOI: https://doi.org/10.7554/eLife.36409.018

## Data availability
Cryo-EM density map of nvTRPM2 has been deposited in the electron microscopy data bank (EMDB) under accession code EMD-7542. Atomic coordinates of nvTRPM2 have been deposited in the protein data bank (PDB) under accession code: 6CO7.

The following datasets were generated:

| Author(s) | Year | Dataset title | Dataset URL | Database, license, and accessibility information |
|---|---|---|---|---|
| Zhe Zhang, Balázs Tóth, Andras Szollosi, Jue Chen, László Csanády | 2018 | Structure of the nvTRPM2 channel in complex with Ca2+ | http://www.rcsb.org/pdb/search/structidSearch.do?structureId=6CO7 | Publicly available at the RCSB Protein Data Bank (accession no. 6CO7) |
| Zhe Zhang, Balázs Tóth, Andras Szollosi, Jue Chen, Lás- | 2018 | Structure of the nvTRPM2 channel in complex with Ca2+ | www.ebi.ac.uk/pdbe/entry/emdb/EMD-7542 | Publicly available at the Electron Microscopy Data |

zló Csanády

Bank (accession no. EMD-7542)

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
