## [Decision Letter]

Thank you for submitting your article "Structure of a TRPM2 channel in complex with Ca^2+^ explains unique gating regulation" for consideration by *eLife*. Your article has been reviewed by three peer reviewers, one of whom, Kenton J Swartz is a member of our Board of Reviewing Editors and the evaluation has been overseen by Kenton Swartz as the Reviewing Editor and Richard Aldrich as the Senior Editor. The following individuals involved in review of your submission have agreed to reveal their identity: Leon D. Islas (Reviewer #2); Alexander Sobolevsky (Reviewer #3).

The reviewers have discussed the reviews with one another and the Reviewing Editor has drafted this decision to help you prepare a revised submission.

Summary:

This is a wonderful manuscript reporting the first near-atomic resolution structure of the TRPM2 channel and a detailed mechanistic study to investigate several of the most intriguing new structural features. The TRPM2 channel is activated by intracellular calcium, ADP-ribose and PIP_2_. Although PIP_2_ is not resolved definitively and the structure was solved in the absence of calcium and ADP-ribose, a calcium binding site near the internal ends of S2 and S3 appears to be identified that is similar to that recently reported for the TRPM4 channel. The external vestibule appears wide enough to pass ions and contains many acidic residues that are important for ion permeation, and the internal pore appears to be closed. One of the more fascinating aspects of the structure is the presence of upper and lower chambers within the cytosolic domain, as well as the tunnels connecting the upper chamber to the putative calcium binding sites and the lower chamber to the bulk solution. The authors show us why the nvTRPM2 channel does not inactivate (unlike the human channel), they show that residues coordinating calcium have either altered calcium affinity or are non-functional, and they present an intriguing phenomenon whereby calcium permeation from outside to inside can reach the calcium binding sites even when the internal surface of the protein is being perfused by calcium buffers. Overall, we think this is one of the nicest TRP channel structures to be reported recently and the work is highly appropriate for *eLife*.

Essential revisions:

1) We are not sure how definitive the authors can be about identification of calcium bound to its putative site. The extensive mutagenesis and comparison with TRPM4 makes us believe that identification of the site is correct, but not whether a calcium ion is actually bound. In subsection “Structure determination” the authors state that the structure was solved in the presence of calcium, but in subsection “Ca^2+^ binding site” they use the word trace. Also, the composition of the solution used to prepare grids is not given. Is it buffer A? We couldn't find the composition of that solution anywhere? It would be nice to see larger density maps in this region with stronger justification. Do you see density for calcium in the half maps? Do you see stronger density for the acidic residues you think are binding calcium compared to other regions of the protein where acidic residues are not involved in hydrogen bonds or ion binding (and are therefore more damaged by the electron beam)? In TRPM4, the authors compared maps with and without calcium to make a stronger case. We are not requesting any new experiments, just expand the presentation/justification and perhaps tone down the conclusion about a calcium ion being present in the maps and models.

2) One of the reviewers thinks that it is almost certain that you have calcium ion accumulation on the internal face of you patches for the experiments shown in Figure 5 and Figure 6. Many of the currents are several nA and these are inside out patches where the membrane will adopt a concave shape towards the bulk solution. This doesn't negate the conclusion that calcium is coming from the outside, but it’s not surprising that calcium can't be washed away at negative voltages when one considers the geometry of the patch and the kinetics of calcium binding to buffers. Please see Stern, 1992, a paper that describes a model that predicts your results for any calcium permeable channel being studied in inside-out patches. According to that model, it might be possible to rule out calcium accumulation using high concentration of a fast buffer such as BAPTA, but you would probably also have to reduce your expression level considerably. You should deal with this point as you see fit, but it would be a shame to misinterpret your results if they have nothing to do with the really beautiful chambers and tunnels.

3) Another reviewer asked why the reactivating current is not as high as that obtained with intracellularly applied Ca^2+^, since one would expect that the entering Ca^2+^ would saturate the binding sites if accessing it through the tunnels. Perhaps intracellular Ca^2+^ induces a separate conformation that allows more efficient coupling to the activation gate?

4) nvTRPM2 represents a highly calcium-permeable TRP channel (PCa/PNa~35). Zhang et al. focus on nvTRPM2 structural features that distinguish its calcium permeability from other TRPMs. However, a brief discussion of how specific these structural features compared to other calcium-permeable channels would broaden the scope of this manuscript and make it more interesting to the general reader. For example, how similar to or different from are the structural mechanisms of high calcium permeability in TRPM2 versus most highly calcium-selective TRP channels TRPV5/6?

---

## [Author Response]

Essential revisions:1) We are not sure how definitive the authors can be about identification of calcium bound to its putative site. The extensive mutagenesis and comparison with TRPM4 makes us believe that identification of the site is correct, but not whether a calcium ion is actually bound. In subsection “Structure determination” the authors state that the structure was solved in the presence of calcium, but in subsection “Ca^2+^ binding site” they use the word trace. Also, the composition of the solution used to prepare grids is not given. Is it buffer A? We couldn't find the composition of that solution anywhere?

The solution used for freezing grids was indeed Buffer A (20 mM Tris-HCl pH 8.0, 150 mM NaCl, 0.06% digitonin, and 1 mM DTT), the same buffer as that used for gel filtration. We now explicitly state that the EM sample was concentrated from the "gel filtration peak fractions in Buffer A" (subsection “EM sample preparation, data collection, and processing”). The buffer was not supplemented with Ca^2+^, so we now state "trace amounts of Ca^2+^" in subsection “Ca^2+^ binding site”. Free Ca^2+^ concentration in a typical saline without added Ca^2+^ or Ca^2+^ buffers is in the μM range (e.g., in an earlier study (Csanády and Adam-Vizi,

2003) we have measured 13 µM), and the standard Vitrobot filter paper we used for freezing grids also contains a certain amount of Ca^2+^, as people have measured (unpublished). Therefore, given the apparent K_1/2_ of ~2 µM for nvTRPM2 (Figure 4E), it is not surprising that bound Ca^2+^ ions are present in our EM map.

It would be nice to see larger density maps in this region with stronger justification. Do you see density for calcium in the half maps? Do you see stronger density for the acidic residues you think are binding calcium compared to other regions of the protein where acidic residues are not involved in hydrogen bonds or ion binding (and are therefore more damaged by the electron beam)? In TRPM4, the authors compared maps with and without calcium to make a stronger case. We are not requesting any new experiments, just expand the presentation/justification and perhaps tone down the conclusion about a calcium ion being present in the maps and models.

To allow better judgement, we now show larger density maps (full map and two half maps) for the region around the Ca^2+^ binding site (Figure 4—figure supplement 1). In the full map and both half maps, the Ca^2+^ densities are all very clear, and the densities for the acidic amino acids which participate in Ca^2+^ binding (D921 and E893) are indeed much stronger than those of adjacent residues that are not involved in hydrogen bonding or Ca^2+^ coordination (E892, D915, and D1106).

2) One of the reviewers thinks that it is almost certain that you have calcium ion accumulation on the internal face of you patches for the experiments shown in Figure 5 and Figure 6. Many of the currents are several nA and these are inside out patches where the membrane will adopt a concave shape towards the bulk solution. This doesn't negate the conclusion that calcium is coming from the outside, but it’s not surprising that calcium can't be washed away at negative voltages when one considers the geometry of the patch and the kinetics of calcium binding to buffers. Please see Stern, 1992, a paper that describes a model that predicts your results for any calcium permeable channel being studied in inside-out patches. According to that model, it might be possible to rule out calcium accumulation using high concentration of a fast buffer such as BAPTA, but you would probably also have to reduce your expression level considerably. You should deal with this point as you see fit, but it would be a shame to misinterpret your results if they have nothing to do with the really beautiful chambers and tunnels.

This is a valid concern that we are fully aware of. We did observe the phenomenon depicted in Figure 5E and Figure 6 even in patches with much smaller currents (residual currents of some tens of pA). Unfortunately, high concentrations of BAPTA, as suggested in the Stern paper, cannot be tested here, because the inevitable introduction of four cations (e.g., Na^+^, K^+^, NMDG^+^) per BAPTA molecule (at pH=7.1) into the bath solution would drastically alter unitary conductance by causing either reversal potential shift or pore block. Thus, we acknowledge that we cannot fully exclude the possibility of some Ca^2+^ accumulation at the cytosolic patch surface, but we also provide four arguments which make this unlikely (subsection “Cytoplasmic cavities and tunnels and access to the Ca^2+^ binding site”):

First, based on the unitary conductances measured with Na^+^ or Ca^2+^ as the charge carrier (Figure 3F-G), when the extracellular solution contains 140 mM Na^+^ and 1 mM Ca^2+^ (Figure 5E, Figure 6), the majority (~74%) of inward nvTRPM2 currents are carried by Na^+^ ions (which account for ~85% of entering cations).

Second, in addition to EGTA, our cytosolic solution contains 140 mM gluconate. Although gluconate is a low-affinity buffer, at 140 mM it binds ~88% of total cytosolic Ca^2+^ (Csanády and Torocsik, 2009), and its high concentration precludes its saturation.

Third, Stern's calculation applies to a static environment (like the cytosol of an intact cell during whole-cell recording) in which buffering around the pore is limited partly by the kinetics of Ca^2+^ binding to the buffer, and partly by the rate of diffusion of the buffer ligand, because the buffer molecules in the vicinity of the pore become rapidly saturated and must be replaced by unbound buffer molecules via diffusion. Here the cytosolic face of the patch is immersed into a rapidly flowing solution. Considering the flow rate of ~1 cm/s of our bath solution and the ~10 nm diameter of a channel protein (Figure 2C), the entire solution volume in contact with the surface of a channel is displaced by fresh Ca^2+^-free solution every ~1 μs, during which time ~2 Ca^2+^ ions enter through its pore (assuming a Ca^2+^ current of 0.6 pA per open channel). Thus, the vectorial flushing effect is likely more important here than diffusion or the kinetics of Ca^2+^ binding to buffers – unless the shape of our patches is concave towards the bulk solution to an extent that severely limits access of the flowing bath solution to the patch surface.

Fourth, the perhaps most compelling argument against Ca^2+^ accumulation at the patch surface being responsible for the residual currents in Figure 5E, (yellow arrows) is our finding that at negative voltages, in the presence of external Ca^2+^, nvTRPM2 currents activate, albeit with a delay, upon addition of cytosolic ADPR even in the absence of cytosolic Ca^2+^ (Figure 6B, right): the magnitude of that "spontaneous" current is identical to that which survives removal of pre-applied cytosolic Ca^2+^ (Figure 6B, left). Because the spontaneous current activates from a state in which all pores in the patch have been closed for several seconds (Figure 6A-B, section (e)), a duration more than sufficient to completely wash off even very high concentrations of pre-applied cytosolic Ca^2+^ (cf., Figure 1A), it cannot be explained by bulk Ca^2+^ accumulation. Rather, its delayed activation must reflect a very low "spontaneous" channel opening rate of ADPR-bound channels in the absence of bound Ca^2+^ (Figure 6A, section (f)): upon the first such spontaneous opening of a channel its upper chamber is immediately flooded by Ca^2+^ which henceforth maintains that channel in an active state (Figure 6A, section (g)).

3) Another reviewer asked why the reactivating current is not as high as that obtained with intracellularly applied Ca^2+^, since one would expect that the entering Ca^2+^ would saturate the binding sites if accessing it through the tunnels. Perhaps intracellular Ca^2+^ induces a separate conformation that allows more efficient coupling to the activation gate?

During "spontaneous" channel activity, with Ca^2+^ entering only through the pore but cytosolic bulk Ca^2+^ concentration kept close to zero, Ca^2+^ concentration in the upper chamber of each channel fluctuates between a very high (during open events) and a low but non-zero value (during closed events). This is because once the gate has closed, Ca^2+^ ions start to dissipate from the chamber through the various windows and tunnels. Thus, whereas all four Ca^2+^ sites of a channel likely become saturated during each open event, some will lose Ca^2+^ during the subsequent closed event. Because channel opening rate is a property of the closed channel, the current that survives cytosolic Ca^2+^ removal (Figure 5E, yellow arrows), or the spontaneous reactivating current (Figure 6, right), reflects channel opening rate under sub-saturating conditions. This explains the lower open probability compared to that seen in the presence of saturating cytosolic Ca^2+^. We have added this explanation to the text (subsection “Cytoplasmic cavities and tunnels and access to the Ca^2+^ binding site”)

4) nvTRPM2 represents a highly calcium-permeable TRP channel (PCa/PNa~35). Zhang et al. focus on nvTRPM2 structural features that distinguish its calcium permeability from other TRPMs. However, a brief discussion of how specific these structural features compared to other calcium-permeable channels would broaden the scope of this manuscript and make it more interesting to the general reader. For example, how similar to or different from are the structural mechanisms of high calcium permeability in TRPM2 versus most highly calcium-selective TRP channels TRPV5/6?

This is an interesting point which we now address in the text (subsection “Ion channel pore”). Although sequence conservation between nvTRPM2 and TRPV6 in the filter and outer vestibule region is low, for both channels the outermost filter position is formed by a ring of four carboxylates (aspartates in TRPV6, glutamates in nvTRPM2), and additional 3-4 negatively charged side chains per subunit are present in the external vestibule. In TRPV6 the latter were termed "recruitment sites" (Saotome et al., 2016). Indeed, closer inspection of nvTRPM2 unitary current sizes (Figure 3G) indicates that the Ca^2+^ throughput rate of the nvTRPM2 pore exceeds that expected for a diffusion-limited channel (Hille et al., 1992). This is a clear indication that local Ca^2+^ concentration in the nvTRPM2 outer vestibule is far higher than that in the bulk solution, suggesting that the negative charges in the nvTRPM2 outer vestibule also act as "recruitment sites".